# DIFFERENTIALLY PRIVATE MECHANISM DESIGN VIA QUANTILE ESTIMATION

**Yuanyuan Yang**
University of Washington
yyangh@cs.washington.edu

**Tao Xiao**
Shanghai Jiao Tong University
xt_1992@sjtu.edu.cn

**Bhuvesh Kumar** *
Snap Inc.
bhuvesh@snap.com

**Jamie Morgenstern**
University of Washington
jamiemmt@cs.washington.edu

## ABSTRACT

We investigate the problem of designing differentially private (DP), revenue-maximizing single item auction. Specifically, we consider broadly applicable settings in mechanism design where agents' valuation distributions are *independent*, *non-identical*, and can be either *bounded* or *unbounded*. Our goal is to design such auctions with *pure*, i.e., $(\epsilon, 0)$ privacy in polynomial time.

In this paper, we propose two computationally efficient auction learning framework that achieves *pure* privacy under bounded and unbounded distribution settings. These frameworks reduces the problem of privately releasing a revenue-maximizing auction to the private estimation of pre-specified quantiles. Our solutions increase the running time by polylog factors compared to the non-private version. As an application, we show how to extend our results to the multi-round online auction setting with non-myopic bidders. To our best knowledge, this paper is the first to efficiently deliver a Myerson auction with *pure* privacy and near-optimal revenue, and the first to provide such auctions for *unbounded* distributions.

## 1 INTRODUCTION

Though prior-dependent auctions, which adjust parameters based on samples of value distributions, often yield better revenue than prior-independent auctions, they risk leaking information about the bids they were trained upon. To address this issue, differential privacy (DP) offers a promising solution (Dwork, 2006; 2008; McSherry and Talwar, 2007; Pai and Roth, 2013), ensuring that a single data point minimally affects the algorithm's output, thus preventing inference of a specific data point.

We study the problem of learning a single-item auction with near-optimal revenue from samples of independent and non-identical value distributions. In this context, the optimal auction (i.e., Myerson's auction (Myerson, 1979)), which relies on value distributions (i.e., prior-dependent), achieves optimal revenue. However, releasing the learned Myerson's auction raises privacy concerns, as the output mechanism may inadvertently reveal sensitive information about the distributions. To *provably* mitigate this risk, our goal is to integrate *pure* DP into the learning process of such auction.

**Pure Differential Privacy**. Given two datasets that differ in one data point, i.e., $D, D'$, we say an algorithm $\mathcal{A}$ satisfies $(\epsilon, \delta)$-*approximate* DP if for any given output $s$: $\Pr[\mathcal{A}(D) = s] \leq e^\epsilon [\mathcal{A}(D') = s] + \delta$. We say $\mathcal{A}$ satisfies *pure* DP if $\delta = 0$. Pure DP allows no slack in privacy protection, and hence is more challenging to achieve than approximate DP. Previous attempts (McSherry and Talwar, 2007; Nissim et al., 2012) to integrate DP with prior-dependent auctions are computationally inefficient or guarantee approximate rather than pure DP. To our knowledge, *no algorithm guarantees pure DP for Myerson's auction in polynomial time*.

**Efficiency**. Incorporating DP into the mechanism often sacrifices efficiency, as achieving privacy guarantees typically incurs additional computational overhead (e.g., random noise addition or extra sampling procedure). In our context, to achieve pure DP, implementing exponential mechanism over

---

*Work done at Georgia Institute of Technology.

all possible mechanisms would incur *exponential* time (See Appendix D). To obtain pure DP more efficiently, we apply recent advances (Durfee, 2023; Kaplan et al., 2022) in private quantile estimation. Our algorithm's running time increases by only *polylog* factors compared to the non-private version.

**Notations** We use $M_A$ to denote the optimal mechanism of distribution $A$, and we use $\text{Rev}(M, A)$ to denote the revenue of deploying mechanism $M$ to distribution $A$. We restricted ourselves to single item auctions; hence, $M_A$ denotes the Myerson auction fitted on distribution $A$, and we denote $\text{OPT}(A) := \text{Rev}(M_A, A)$ as the optimal revenue one could get from a distribution $A$. We use $\mathbf{1}_k$ to denote a $k$-dimensional vector with all entries equal to 1. We use $\widetilde{O}$ and $\widetilde{\Theta}$ to hide polylog factors.

## 1.1 RESULTS

Formally, we define the problem of learning a near-optimal auction with a pure DP:

**Problem 1.1** (Optimal Auction with $(\epsilon_p, 0)$-DP). Given $n$ samples of $k$-dimensional distribution $\mathbf{D}$, the goal is to learn a single item auction $M$ with $(\epsilon_p, 0)$-DP, whose expected revenue on $\mathbf{D}$ is close to the optimal revenue, i.e., with prob. $1 - \delta^1$, $|\mathbb{E}[\text{Rev}(M, \mathbf{D}) - \text{OPT}(\mathbf{D})]| \leq \epsilon$ for some small $\epsilon$.

**Insight**. To address this problem, we leverage the insight that, the expected optimal revenue from value distribution is *insensitive* to small statistical shifts and discretization in the quantile and value space. Additionally, we observe that the accuracy of the points returned by private quantile estimation (QE), assuming the data points follow a distribution, directly correlates with the statistical distance between the distribution formed by the returned points and the true distribution. Thus, we can reduce private Myerson fitting from samples to *private quantile estimation of pre-specified quantiles*.

Achieving pure DP while maintaining meaningful revenue guarantees is challenging. A crucial aspect is to ensure that the values (hence distribution) returned by DP Quantile Estimation (QE) possess meaningful and provable accuracy guarantees. To obtain such accuracy, our algorithm (Alg. 1) first additively discretize the empirical distribution in the value space to distribution $\widehat{D}^\epsilon$, then estimate the pre-specified quantiles with DPQE. We improved the accuracy bound of DPQE (DPQUANT,Kaplan et al. (2022)) to accommodate cases with duplicate values. This improved bound allows us to upper bound the statistical distance between the output distribution and $\widehat{D}^\epsilon$, thus upper bounding the revenue loss incurred from fitting a Myerson on the output distribution.

Theorem 1.2 briefly presents the near-optimal revenue of our proposed mechanism. The final privacy parameter has a dependency on $k$ since the output of mechanism $M$ is of dimension $2k$. We present complete details in Section 3 and the complete theorem statement in Theorems 3.2 and 3.3.

**Theorem 1.2** (Revenue Guarantee of Private Myerson, Bounded). *Given $n = \widetilde{\Theta}(\epsilon^{-2})$ samples $\widehat{V}$ of the joint distribution $\mathbf{D} \in [0, h]^k$, there exist a mechanism $M$ that is $2k\epsilon_p$ differentially private with running time $\widetilde{\Theta}(kn)$ and takes $\widetilde{\Theta}(1)$ pass of the distribution. With probability $1 - \delta$, this mechanism $M$ satisfies:$|\mathbb{E}[\text{Rev}(M, \mathbf{D}) - \text{OPT}(\mathbf{D}))]| \leq \widetilde{O}((\epsilon + \epsilon^2/\epsilon_p)kh)$.*

The prior algorithm does not work for *unbounded* distributions. Our second algorithm (Alg. 9) addresses the case for $\eta$-*strongly regular* value distributions by efficiently truncating them to bounded distributions with small expected revenue loss. This approach enables the application of our previous mechanism (Alg. 1) designed for the bounded distribution case. Since the truncation point is a function of the optimal revenue, we develop Alg. 7 to approximate this point by achieving a $\widetilde{\Theta}(k)$-approximation of the optimal revenue, where $k$ denotes the dimension of the product distribution.

Theorem 1.3 outlines the accuracy of our proposed mechanism for certain parameter settings. Since this truncation point depends adaptively on the desired accuracy, the revenue gap exceeds that for the bounded case, and the tradeoff between privacy and revenue are more pronounced. We present more details in Section 4, and the complete theorem statement is in Theorems 4.1 and I.13.

**Theorem 1.3** (Revenue Guarantee of Private Myerson, Unbounded). *Given $n = \widetilde{\Theta}(\epsilon^{-2})$ samples $\widehat{V}$ of $\eta$-strongly regular joint distribution $\mathbf{D} \in \mathbb{R}^k$, there exist a mechanism $M$ for unbounded distribution that is $2k\epsilon_p$ differentially private with running time $\widetilde{\Theta}(kn)$ and takes $O(n)$ passes. With probability $1 - \delta$, this mechanism $M$ satisfies: $|\mathbb{E}[\text{Rev}(M, \mathbf{D}) - \text{Rev}(M_\mathbf{D}, \mathbf{D})]| \leq \widetilde{O}(k^2\sqrt{\epsilon} + k^2\epsilon^{1.5}/\epsilon_p)$.*

---

[1]This failure probability $\delta$ is inevitable due to the inherent uncertainty in learning from a finite sample set, see Chapter 1 Kearns and Vazirani (1994)

**Application: Online auction with nonmyopic bidders**. We now describe how our mechanisms incentivize truthful bidding from nonmyopic bidders under practical online auction settings.[2] In the online setting, auctions are deployed iteratively and later auctions are informed by previous bids. Since future auctions can be affected by earlier bids, *nonmyopic* bidders may strategically bid in earlier rounds to increase winning chances and/or secure lower prices, increasing their utility.

To prevent from strategic bidding, we integrate our previous solutions (Alg. 1, Alg. 9) with a commitment mechanism. Our DP Myerson naturally upper bound the utility gain (of future rounds) by definition, in that the change of one bid affect the outcome's probability by privacy parameter $\epsilon_p$. Our algorithm operates in two stages. In the first stage, it employs a commitment mechanism that penalizes strategic bids. In the second stage, the algorithm fits a DP Myerson auction from the collected bids and generates revenue in the remaining rounds. This approach ensures that strategic bids only lies in a small neighbor of the true value; otherwise, the bidder's utility becomes negative.

We present the *regret* (i.e., the time-averaged revenue of the proposed mechanism compared to the optimal one) of our proposed mechanism (Alg. 3) in Theorem 1.4, which shows the accuracy of our algorithm in terms of regret. We defer readers to Section 5 and Theorem 5.4 for further details.

**Theorem 1.4** (Revenue Guarantee of Online Mechanism). *Given $\epsilon \in [0, 1/4]$, under the online auction setting described in Section 5.1), there exists an algorithm (Alg. 3) run with parameter $T = \widetilde{\Theta}(\epsilon^{-2})$ that, with probability $1 - \delta$, achieves diminishing regret, i.e.,* REGRET $= \widetilde{O}[(\epsilon + \sqrt{\eta\epsilon})kh]$, *where $\eta$ is a constant specific to bidders' utility model.*

## 1.2 PRIOR WORK

**DP Mechanism Design**. Emerging from McSherry and Talwar (2007), there has been interest in delivering mechanisms with DP guarantees (Nissim et al., 2012; Huang et al., 2018a; Zhang and Zhong, 2022; Huh and Kandasamy, 2024). These mechanism are either *no longer optimal* in our setting, or doen't generalize to unbounded distribution setting.

**Online Learning in Repeated Auction**. Regarding the single item online auction setting, Kanoria and Nazerzadeh (2014); Huang et al. (2018a) established near-optimal solutions when bidders' utility is discounted and valuations are i.i.d.. Deng et al. (2020); Abernethy et al. (2019) introduced specific incentive metrics to quantify bidders' willingness to bid other than their true values and developed mechanisms that minimize incentives for strategic bidding under these metrics in large markets.

For a detailed, complete list of related work topics, please see Appendix C.

## 1.3 CONTRIBUTIONS

**Revenue Maximizing Auctions with Pure Privacy Guarantee**. Our work is the first to develop a mechanism with *pure* DP that obtains near optimal revenue for single item auction with independent and non-identical bidders, and for both *bounded* and *unbounded* $\eta$-strongly regular distributions. For bounded distributions, our mechanism achieves optimal time complexity within polylog factors.

**Application to Online Auction Setting**. We apply our mechanism into the online auction setting with nonmyopic, independent and non-identical bidders. Combined with our designed commitment strategy, the integrated solution restricts the bids to a small neighbor around the corresponding value. Consequently, these approximately truthful bids enables our solution to generate revenue guarantee that converges to the optimal revenue over time, for time-discounted, or large market bidders. We generalize the i.i.d bidder setting in Huang et al. (2018a) and solve the open problem they proposed.

**Extended Analysis of Private Quantile Algorithm**. We extend the analysis of the quantile estimation oracles employed in this paper. For quantile estimation on bounded datasets (Kaplan et al., 2022), the paper assumes that all data points are *distinct* and derive accuracy bounds dependent on the dataset's range. We generalize their analysis to accommodate cases where multiple data points may share *identical* values. Additionally, for quantile estimation of unbounded distributions (Durfee, 2023), we provide theoretical accuracy guarantees, complementing the paper's focus on empirical performance.

---

[2]Recognizable non-i.i.d. value distributions are common, e.g., Meta Ad platform (met) requires that each advertiser selects one of six objectives, corresponding to different distributions based on the industry or ad topic.

## 2 PRELIMINARIES

In this section, we outline the preliminaries on mechanism design, differential privacy, and quantile estimation. Additional information can be found in Appendix E.

### 2.1 MECHANISM DESIGN BASICS

We now formally define the allocation rule and payment rule of a single item auction.

**Definition 2.1** (Allocation Rule and Payment Rule). Given $k$ bidders with bid $\mathbf{b} := (b_1, \ldots, b_k)$, a single-item auction $M$ consists of an allocation rule as $\mathbf{x}(\mathbf{b}) := (x_1(\mathbf{b}), \ldots, x_k(\mathbf{b})) \in [0,1]^k$ and a payment rule as $\mathbf{p}(\mathbf{b}) := (p_1(\mathbf{b}), \ldots, p_k(\mathbf{b})) \in [0,1]^k$, where $x_j$ denotes the probability that the $j$-th bidder gets the item, and $p_j$ denotes her payment.

Under truthful sample access, the Myerson's auction maximizes the expected revenue.

**Definition 2.2** (Myerson's Single Item Auction (Myerson, 1981)). For a discrete product distribution $\mathbf{D} = \mathcal{D}_1 \times \ldots \times \mathcal{D}_k$ (Elkind, 2007), the *virtual value* for $\mathcal{D}_j$ at value $v_i^j$ with support $\mathcal{V}_j = \{v_1^j, \ldots, v_n^j\}$ is $\phi_j(v_i^j) = v_i^j - (v_{i+1}^j - v_i^j)\frac{1-F_j(v_i^j)}{f_j(v_i^j)}$, where $v_i^j$s are ordered in increasing order of $i$, $f_j(v_i^j) = \mathbb{P}[v^j = v_i^j]$, and $F_j(v_i^j) = \sum_{k=1}^{i} f(v_k^j)$.

We say the product distribution $\mathbf{D}$ is $\eta$-*strongly regular* if for all $j$, $\phi_j(v_i) - \phi_j(v_j) \geq \eta(v_i - v_j)$ for every $v_i > v_j \in \mathcal{V}$ and $\eta > 0$.

For these distributions $\mathcal{D}$ with nondecreasing virtual value, *Myerson's allocation rule* $x_i(v_i) = \mathbb{1}\{\phi_i(v_i) \geq \max(0, \max_{j \neq i} \phi_j(v_j))\}$, where $\mathbb{1}\{\cdot\}$ denotes the indicator function. The *payment rule* $p_i(v_i) = \mathbb{1}\{\phi_i(v_i) \geq \max(0, \max_{j \neq i} \phi_j(v_j))\}\phi_i^{-1}(\max(0, \max_{j \neq i} \phi_j(v_j)))$. [3]

### 2.2 DIFFERENTIAL PRIVACY BASICS

We present the definition of pure DP and approximate DP below.

**Definition 2.3** (Differential privacy). An algorithm $\mathcal{A} : \mathbb{R}_+^n \to \mathbb{R}$ is $(\epsilon, \delta)$-*approximate* DP if for neighboring dataset $V, V' \in \mathbb{R}_+^n$ that differs in only one data point, and any possible output $O$, we have: $\Pr[\mathcal{A}(V) = O] \leq \exp(\epsilon) \Pr[\mathcal{A}(V') = O] + \delta$. We say it satisfies *pure* DP for $\delta = 0$.

A key property we leverage from differential privacy is its immunity to post-processing. Post-processing refers to any computation or transformation applied to the output of a DP algorithm after the data has been privatized. In our context, Myerson's auction can be seen as a post-processing step. Therefore, applying Myerson's auction to a differentially private release of the empirical distribution preserves the original privacy guarantees of the input distribution.

**Lemma 2.4** (Immunity to Post-Processing). *Let $\mathcal{A} : \mathbb{R}_+^n \to \mathbb{R}$ be an $(\epsilon, \delta)$-DP algorithm, and let $f : \mathbb{R} \to \mathbb{R}$ be a random function. Then, $f \circ \mathcal{A} : \mathbb{R}_+^n \to \mathbb{R}$ is also $(\epsilon, \delta)$-DP.*

### 2.3 QUANTILE ESTIMATION

Quantile estimation (QE) is used for estimating a value of specified quantiles from samples. Given samples from a distribution, an accurate QE from samples directly translates to an accurate CDF estimation of the underlying distribution. Below, we formally introduce the definition of QE.

**Definition 2.5** (Quantile Estimation). Given a range of the data as $H$, a dataset $X \subseteq H^n$ containing $n$ points from range $H$, and a set of $m$ quantiles $0 \leq q_1, \ldots, q_m < 1$, identify quantile estimations $v_1, \ldots v_m$ such that for every $j \in [m]$, $|\{x \in X | x \leq v_j\}| \approx q_j \cdot n$. [4]

We now present the definition of *statistical dominance* and *KS-distance* below.

**Definition 2.6** (Stochastic Dominance and KS-Distance). Given distribution $\mathcal{D}$ and $\mathcal{D}'$, we denote the CDF of them as $F_\mathcal{D}, F_{\mathcal{D}'}$, respectively. Distribution $\mathcal{D}$ stochastically dominates distribution $\mathcal{D}'$ (denoted as $\mathcal{D} \succeq \mathcal{D}'$) if: (1) For any outcome $x$, $F_{\mathcal{D}(x)} \leq F_{\mathcal{D}'}(x)$. (2) For some $x$, $F_{\mathcal{D}(x)} < F_{\mathcal{D}'}(x)$. The KS distance between $\mathcal{D}$ and $\mathcal{D}'$ is $d_{\mathrm{ks}}(\mathcal{D}, \mathcal{D}') = \sup_{x \in \mathbb{R}} |F_{\mathcal{D}(x)} - F_{\mathcal{D}'}(x)|$.

---

[3] We define the virtual value inverse $\phi_i^{-1}(\phi)$ as $\arg\min_{v \in \mathcal{V}} \phi_i(v) \geq \phi$.

[4] More formally, $v_j \in X$ is the minimum value such that this quantity exceeds $q_j n$.

# 3 PRIVATE MYERSON'S AUCTION FOR BOUNDED DISTRIBUTIONS

In this section, we introduce the algorithm for fitting a Myerson's auction with a pure privacy guarantee. To ensure pure privacy, since DP is immune to postprocessing, it is sufficient to input a private distribution estimated from samples to the Myerson. The challenge lies in finding such distributions that still yield near-optimal revenue.

Our approach leverages private quantile estimation (QE) over samples to achieve the desired guarantee. However, the standard guarantees of DPQE collapse when the dataset contains points that are extremely close. This is a critical issue in our setting, as increasing the sample size $n$ from continuous value distributions inherently causes the minimum distance between samples to approach zero. To address this, we introduce additional discretization steps to prevent non-identical points from being too close together, and we develop new DPQE guarantees specifically tailored to handle samples with identical values.

## 3.1 PRIVATE MYERSON FOR BOUNDED DISTRIBUTIONS

Next, we present DPMYER algorithm (Alg. 1). The algorithm first value-discretize the samples of the distribution additively by $\epsilon_a$, then quantile-discretize these samples by $\epsilon_q$ with pure privacy guarantee. Specifically, the quantile discretization estimates the values of the quantile set $[\epsilon_q, 2\epsilon_q, \ldots, 1]$ with pure privacy. Next, DPMYER use the estimated quantile values and the quantile set to construct a distribution, then perturb it to a final distribution that is stochastically dominated by the ground truth. Finally, the final distribution is then used to implement Myerson's mechanism.

---

**Algorithm 1** DP Myerson, Bounded Distribution DPMYER$(V, \epsilon_q, \epsilon_a, h, \epsilon_p)$

---

**Input:** $n$ samples $V \in R_+^{k \times n}$, discretization parameter $\epsilon_q, \epsilon_a$, upper bound $h$, privacy parameter $\epsilon_p$

1: Discretize all values into multiples of $\epsilon_a$; let the resulting samples be $\widehat{V}$.
2: Prepare the quantile to be estimated: $Q \leftarrow \{\epsilon_q, 2\epsilon_q, \ldots, \ldots, \lfloor (1/\epsilon_q) \rfloor \cdot \epsilon_q, 1\}$.
3: For each dimension $i \in [k]$, decide the prices $\widehat{S}_{[i,:]} \leftarrow \text{QESTIMATE}(Q, V_{[i,:]}, \epsilon_p)$.
4:                                       ▷ Estimate the quantiles by DPQUANT (Alg. 4)
5: Construct distribution $\widetilde{D}$ based on $\widehat{S}$, treating the valuations in $\widehat{S}$ as if each has probability $\epsilon_q$.
6: For each $i \in [k]$, shift the top $\epsilon_q$ quantile of $\widetilde{D}_i$ to the bottom, fit Myerson on this distribution.

---

## 3.2 REVENUE OPTIMALITY AND RUNNING TIME

Next, we show the revenue optimality and the efficiency of our algorithm. To upper bound the reveue loss, we derive the revenue shift theorem, which upper bounds the revenue difference between two distributions by a linear function of their statistical distance.

**Theorem 3.1** (Revenue Shift Theorem). *Given two product distribution $\mathbf{D} \succeq \mathbf{D}'$ whose valuations are bounded by $h$, with $d_{ks}(\mathbf{D}_i, \mathbf{D}'_i) \leq \alpha_i$ for any bidder $i$, the optimal revenue of these distribution satisfies:* $0 \leq \mathbb{E}[\text{Rev}(M_\mathbf{D}, \mathbf{D}) - \text{Rev}(M_{\mathbf{D}'}, \mathbf{D}')] \leq (\sum_{i \in [k]} \alpha_i)h$.

We apply this theorem to upper bound the revenue loss between 1) the quantile-discretized distribution and its pre-quantized counterpart, and 2) the distribution obtained from private quantile estimation and that from the groundtruth quantile estimation. The first one is evident, while the second arises from DPQUANTILE's ability to control the KS-distance between the estimation and the ground truth.

We now present the accuracy guarantee of the private Myerson algorithm. Provided the privacy parameter is not too small (i.e, $\epsilon_p = \Omega(\epsilon^{-1})$), our guarantee implies that the optimal revenue of the distribution does not exceed the revenue of our algorithm on its samples by more than $\widetilde{\Theta}(\epsilon k h)$.

**Theorem 3.2** (Revenue Guarantee of Private Myerson (Alg. 1)). *Given $n$ samples $\widehat{V} \in [0, h]^{k \times n}$ of the joint distribution $\mathbf{D}$, DPMYER (Alg. 1) is $(2k\epsilon_p, 0)$-DP, and the expected revenue of this mechanism is close to the optimal revenue of distribution $\mathbf{D}$, i.e., with probability $1 - \delta$:*

$$| \mathbb{E}[\text{Rev}(M_{\text{DPMYER}}, \mathbf{D}) - \text{OPT}(\mathbf{D})]| \leq \widetilde{O}((\epsilon + \epsilon^2/\epsilon_p)kh).$$

*under parameter $\epsilon_a = \epsilon_q = \epsilon$ and $n = \widetilde{\Theta}(\epsilon^{-2})$, where we hide the polylog factors in $\widetilde{\Theta}$ and $\widetilde{O}$.*

*Proof Sketch.* We begin by deriving the privacy guarantee of our algorithm. Next, we establish an upper bound on the distance between the private distribution $\widehat{D}^p$ and the additively discretized distribution $\widehat{D}^\epsilon$. This enables us to apply the revenue shift theorem (Thm.3.1) to upper bound the revenue loss from private quantile estimation. By aggregating this loss with the revenue loss due to value discretization, we arrive at the final result. In this proof sketch, we omit the polylog factors that depends on $k, n, \delta, \epsilon_a, \epsilon_p, \epsilon_q$ for a clear presentation. Further details are provided in Appendix H.2.

**Privacy Guarantee.** We know that the quantile estimates from DPQE is $(\epsilon_p, 0)$ private (Lem. H.2). Since DP is immune to post-processing (Lem. E.4), and that the output of allocation and payment combination is $2k$ dimensional, by composition theorem (Lem. E.5), our algorithm is $(2k\epsilon_p, 0)$-DP.

**Upper Bounding the Statistical Distance** The distribution $\widehat{D}^p$ is obtained by changing from distribution $\mathbf{D}$ through distribution $\widehat{D}$, the distribution $\widehat{D}^\epsilon$ and $\widehat{D}^q$ (Figure 1). We upper bound the statistical KS distance of these distributions: 1) By DKW inequality, we upper bound the KS-distance between $\widehat{D}$ and $\mathbf{D}$ by $\widetilde{\Theta}(1/\sqrt{n})$ for each coordinate $i$ (with probability $1 - \delta/2$). 2) By definition, we upper bound the KS-distance between $\widehat{D}^\epsilon$ and $\widehat{D}^q$ by $k\epsilon_q$. 3) By developing and converting the bound of the DP quantile algorithm (Lem. H.3) into a bound on the CDF, we upper bound the KS-distance between $\widehat{D}^q$ and $\widehat{D}^p$ by $k\widehat{\epsilon}$ for $\widehat{\epsilon} := \widetilde{\Theta}(1/(\epsilon_p n))$ (with probability $1 - \delta/2$).

**Upper Bounding the Revenue Loss.** We then upper bound optimal revenue loss from $\mathbf{D}$ to $\widehat{D}^p$. This upper bound can be obtained by combining the revenue loss from the aforementioned distributions (by revenue shift theorem), with an additive $\epsilon_a$ revenue loss from discretization (by Lem. F.1). The revenue loss from statistical shift aggregates to $\widetilde{\Theta}((1/\sqrt{n} + \epsilon_q + \widehat{\epsilon})kh)$ with probability $1 - \delta$.

**Putting it all together.** Finally, condition on the DPQUANT proceeds successfully and the samples are close to the underlying distribution (with probability $1 - \delta$), we get that the expected revenue of DPQUANT on the underlying distribution is at least the optimal revenue from this distribution minus the revenue difference between $\mathbf{D}$ and $\widehat{D}^p$ by the following inequality:

$$0 \geq \mathbb{E}[\mathsf{Rev}(M_{\widehat{D}^p}, \mathbf{D}) - \mathsf{OPT}(\mathbf{D})] \geq \mathbb{E}[\mathsf{Rev}(M_{\widehat{D}^p}, \mathbf{D}) - \mathsf{OPT}(\widehat{D}^p)] - |\mathsf{OPT}(\widehat{D}^p) - \mathsf{OPT}(\mathbf{D})|$$

where the first inequality follows from the optimality of $M_\mathbf{D}$ on $\mathbf{D}$ and the second inequality follows from adding $\mathsf{OPT}(\widehat{D}^p)$. By our construction of $\widehat{D}^p$, this distribution is stochastically dominated by $\mathbf{D}$, thus from the strong revenue monotonicity (Lem. F.3), we get that $\mathbb{E}[\mathsf{Rev}(M_{\widehat{D}^p}, \mathbf{D}) - \mathsf{OPT}(\widehat{D}^p)] \geq 0$. Thus, we concluded that the revenue gap is upper bounded by $\widetilde{\Theta}((1/\sqrt{n} + \epsilon_q + \widehat{\epsilon})kh + \epsilon_a)$. We set $\delta$ in the statement as $1/k$ of the $\delta$ we used in this proof to generate the final revenue guarantee. $\square$

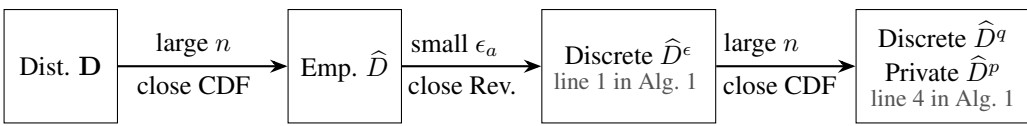

Figure 1: **Distribution analyzed for DPMYER(Alg. 1)**. We establish connections between the accuracy/revenue guarantee of the original distribution $\mathbf{D}$ with the empirical distribution $\widehat{D}$, the value-discretized $\widehat{D}^\epsilon$, the quantile-discretized $\widehat{D}^q$ and the distribution $\widehat{D}^p$ returned by DPQUANT(Alg. 4).

Next, we demonstrate the efficiency of our algorithm, which is achieved through a organized implementation of the DP Quantile algorithm. Intuitively, given $m$ ordered quantiles, the algorithm iteratively identifies and estimates the median (the $m/2$-th), followed by the $m/4$ and the $3m/4$ quantiles, and so on. This hierarchical structure ensures that each data point is used in at most $\log m$ quantile estimates (of a single quantile). For more details, we refer readers to Appendix H.1.

**Theorem 3.3** (Time Complexity for Private Myerson, Bounded)**.** *Given the same parameters as stated in Theorem 3.2,* DPMYER *(Alg.1) runs in* $\widetilde{\Theta}(kn)$ *time and requires* $\widetilde{\Theta}(1)$ *passes of the samples.*

*Proof Sketch.* The time dominant step is *quantile estimation*, which requires $\log(\lfloor 1/\epsilon_q \rfloor + 1)$ passes of the dataset. It takes $O(k \log(\lfloor 1/\epsilon_q \rfloor + 1)/(\epsilon_a \epsilon_q)) = \widetilde{\Theta}(kn)$ time, since $n = \widetilde{\Theta}(\epsilon^{-2})$. This step calculates the utility of $k \lfloor h/\epsilon_a \rfloor$ over $\lfloor 1/\epsilon_q \rfloor$ quantiles for at most $\widetilde{\Theta}(1)$ time. For full version of this proof, please refer to Appendix H.3 $\square$

## 4 GENERALIZATION TO UNBOUNDED DISTRIBUTIONS

Generalizing the DP Myerson mechanism to unbounded distributions introduces new challenges. The revenue loss upper bound produced by previously introduced *quantile estimation* algorithm and *revenue shift* theorem both depends (positively) on the range of the distribution. Without a finite range, these upper bound becomes infinite and fail to effectively control the revenue loss.

We consider the widely accepted $\eta$-strongly regular distributions, which decays at least as fast as exponential distributions. A key element of our approach is appropriately truncating the distribution, which enables us to extend the discretize-then-DP-quantile method to the unbounded setting. Specifically, we apply the property of the regular distribution that (Devanur et al., 2016), truncating the distribution by $\frac{1}{\epsilon}$OPT($\mathbf{D}$) costs at most $2\epsilon$ fraction of the optimal revenue (Lem. I.1). Hence, for the truncation to work, it is essential to approximate the optimal revenue based on sample data. Meanwhile, incorporating the truncation with pure DP introduces additional complexities.

We are now ready to present our approach for a $k$-approximation of the optimal revenue with pure DP for $\eta$-strongly regular product distributions. Our DPKOPT (Alg. 2) algorithm approximates the optimal revenue by running a empirical reserve(ER) over *each* bidder's distribution truncated at the top $\eta^{1/(1-\eta)}/4$ quantile.[5] Summing up these estimates gives us a $\Theta(k)$-approximation of the optimal revenue, by the fact that $k$OPT($\mathbf{D}$) $\geq \sum_{i\in[k]}$ OPT($\mathcal{D}_k$) $\geq$ OPT($\mathbf{D}$).

---

**Algorithm 2** DP Estimation for Optimal Revenue DPKOPT($V, \epsilon_q, \epsilon_a, \epsilon_p, \eta$)

---

**Input:** $n$ samples $V = \{\mathbf{v}_1, \ldots, \mathbf{v}_n\}$, quantile discretization $\epsilon_q$, additive discetization $\epsilon_a$, privacy parameter $\epsilon_p$, regularity parameter $\eta$.

1: **for** $d = 1 \rightarrow k$ **do**
2:     $\widehat{q} \leftarrow 1/4 \cdot \eta^{1/1-\eta}$
3:     Let $ub_d \leftarrow$ DPQUANTU($V_{[d,:]}, 1 - \widehat{q}$).           ▷ Estimate the truncation point of $D_d$.
4:     Truncate distribution $D_d$ at $ub_d$ as $\widehat{D}_d$, and discretize $\widehat{D}_d$ by additive $\epsilon_a$ in the value space.
5:     Prepare the quantile to be estimated, $Q \leftarrow \{1 - \widehat{q}, 1 - \widehat{q} - \epsilon_q, \cdots, 1 - \widehat{q} - \lfloor\frac{1-\widehat{q}}{\epsilon_q}\rfloor \cdot \epsilon_q, 0\}$.
6:     $\widehat{S}_{[d,:]} \leftarrow$ QESTIMATE($Q, V_{[d,:]}, \epsilon_p$)           ▷ Apply DP quantile estimate (Alg. 4).
7:     Let $\widehat{F}_d$ be the distribution generated by value profile $\widehat{S}_{[d,:]}$ and quantile set $Q$.
8:     SREV$_d \leftarrow \max_{r \in \widehat{S}} r(1 - \widehat{F}_d(r))$.       ▷ Estimate the optimal revenue from $\widehat{F}_d$ (Alg. 6).
9: **end for**
10: KREV $\leftarrow \sum_{d \in [k]}$ SREV
11: **return** KREV

---

To guarantee pure privacy, our algorithm estimates the optimal revenue using a DP-estimated proxy $\widehat{F}_{[k]}$ derived from the sample data. This proxy is obtained from truncating the distribution by DPQUANTU (Alg. 7) and quantile-discretizing the distribution by DPQUANT. During this process, the truncation by DPQUANTU cost at most a constant fraction of the optimal revenue, and DPQUANT cost at most an additional $\widetilde{\Theta}(\frac{1}{\epsilon_p n}k + \epsilon_a)$. Aggregating these revenue loss concludes that the output is a $\Theta(k)$-approximation of the optimal revenue. See Appendix I.4 for more details.

Our private Myerson algorithm for the unbounded distribution (Alg. 9) integrate DPKOPT and yields the following accuracy bound. See Appendix I.5 for formal statements and more details.

**Theorem 4.1** (Revenue Guarantee of Private Myerson, Unbounded). *Given* $\epsilon \in [0, 1/4]$, $n$ *samples* $\widehat{V}$ *of the joint distribution* $\mathbf{D} \in [0, h]^k$, *the output of Myerson fitted under* DPMYERU *(Alg. 9) is* $(2k\epsilon_p, 0)$-*DP, and under* $\epsilon_a = \epsilon_q = \epsilon$, $n = \widetilde{\Theta}(\epsilon^2)$, $n_1 = \widetilde{\Theta}(\epsilon^2)$, $\epsilon_t = \sqrt{\epsilon}$, *with probability* $1 - \delta$,

$$\mathbb{E}[\mathsf{Rev}(M_{\mathrm{DPMYERU}}, \mathbf{D}) - \mathsf{Rev}(M_{\mathbf{D}}, \mathbf{D})]| \leq \widetilde{O}(k^2\sqrt{\epsilon} + k^2\epsilon^{1.5}/\epsilon_p)$$

---

[5]Without privacy constraints, truncating at the top $\eta^{1/(1-\eta)}$-suffices by Lem. I.4. Our algorithm adopt a looser truncation since the DPQUANTU algorithm only return the value of given quantiles *approximately*.

# 5 APPLICATION: ONLINE MECHANISM DESIGN FROM BIDS

We now study how to integrate our previous solutions into the online auction setting, such that, the algorithm produces time-averaged revenue guarantee that converges to the optimal. The auction now spans multiple rounds, where each auction is informed by the bids from previous rounds. We consider the setting where bidders are non-myopic bidders, and have incentives to bid strategically in the current round to increase their utilities over future auctions.

## 5.1 APPLICATION BACKGROUND

Before presenting our algorithm, we first provide the formal problem definition of the online auction setting. We study online mechanism design over a time horizon of $T$, where an identical item is sold at each iteration. Each bidder has a *publicly observable* attribute. Bidders with the same attribute have the same valuation distribution.

We are now ready to describe interactions between bidders and the auctioneer over time horizon $T$, as shown in Figure 2. We defer to Appendix J.2 for more details how bidder generates the samples.

---

For each time $t \in [T]$:

- The learner/auctioneer sells a fresh copy of the item.
- The learner collects the bids in the form of $(b_j, a_j)$, where $b_j$ and $a_j$ denote the bid and the attribute of the $j \in [d_t]$-th bidder, respectively.
- The learner decides the allocation rule $\mathbf{x}_t$ and payment $\mathbf{p}_t$ accordingly.

---

Figure 2: Online Auction with $k$ Attributes.

Each item the auctioneer sells is identical, and each bidder has an additive (discounted) utility of the items across rounds. We consider the bidders either have *discounted utility* or are in a *large market*.

**Definition 5.1** (Bidder's Utility). Each bidder $j$ has a quasi-linear utility function at time $t$: $u_j^t = x_j^t(v_j^t - p_j^t)$, where $x_j^t, v_j^t, p_j^t$ are the allocation, value, price for bidder $j$ at time $t$, respectively. We consider two *nonmypoic* bidders' utility models:

*Discounted Utility*: For discount factor $\gamma \in [0, 1]$, the bidders seek to maximize the sum of utilities discounted by $\gamma$. At the $t$-th iteration, the discounted utility is $\widehat{u}_j^t = \sum_{r=t}^T u_j^r \gamma^{r-t}$.

*Large Market*: (Anari et al., 2014; Jalaly Khalilabadi and Tardos, 2018; Chen et al., 2016): The bidder only participates in a subset $S_j$ of auctions, i.e., for each $u^{1:T} = \sum_{t \in S_j} u^t$, with subset $|S_j| < l$.

Ideally, the learner's objective is maximize time-averaged revenue with high probability. Our regret compare this revenue against the optimal revenue of the (unobservable) value history.

**Definition 5.2** (Learner's Objective). Given $\delta$, the learner's objective is to decide an allocation $\mathbf{x}_{1:T}$ and a payment $\mathbf{p}_{1:T}$ that achieves sublinear regret, i.e., with probability $1 - \delta$,

$$\text{REGRET} := \frac{1}{T} \sum_{t \in [T]} \mathbb{E}[\text{Rev}(\mathbf{x}_t, \mathbf{p}_t, \mathbf{b}_t) - \mathbb{E}[\text{OPT}(\mathbf{v}_t)]] = o(1),$$

with the expectation taken over the value distribution.

## 5.2 TWO-STAGE MECHANISM FOR BOUNDED DISTRIBUTION

This two-stage algorithm (Alg. 3) consists of repeated auctions over $T$ rounds, and the participating bidders' values in each round are upper bounded by a *known* constant $h$. The algorithm first collects the samples for the first $T_1$ rounds, by running a commitment algorithm (Alg. 10) that punishes nontruthful bids. Then, the algorithm deploys our previously developed DP Myerson's Algorithm (Alg. 1, Alg. 9) for the remaining rounds to obtain near optimal revenue. In addition to these two steps, our algorithm includes a step where all samples are reduced by $\nu$ (line 4 of Alg. 3) and projected onto nonnegative value spaces. This step is designed to offset the impact of strategic bidding.

---

**Algorithm 3** Two-Stage Algorithm $\mathcal{A}_{\text{BOUNDED}}$

---

**Input:** Rounds $T$, learning rounds $T_1$, parameter $\epsilon_a, \epsilon_q, \epsilon_p, \nu$, upper bound $h$.
1: **for** $t \leftarrow 1, \ldots T_1$ **do**     ▷ Collection Stage
2:     Receive bids $\mathbf{b}^t$, and attributes $\mathbf{a}^t$.
3:     Return $(\mathbf{x}^t, \mathbf{p}^t) \leftarrow \text{COMMIT}(\mathbf{b}^t)$.     ▷ Commitment Algorithm(Alg. 10)
4:     $\widehat{\mathbf{b}}^t \leftarrow \mathbb{P}_{[0,h]}[\mathbf{b}^t - \nu \mathbf{1}_k]$
5: **end for**
6: $(\widetilde{\mathbf{x}}(\cdot), \widetilde{\mathbf{p}}(\cdot)) \leftarrow \text{DPMYER}(\widehat{\mathbf{b}}^{1:T_1}, \mathbf{a}^{1:T_1}, \epsilon_q, \epsilon_a, h, \epsilon_p)$  ▷ Fit Myerson's auction (Alg. 1, or Alg. 9)
7: **for** $t \leftarrow T_1 + 1, \ldots T$ **do**     ▷ Revenue Stage
8:     Receive bids $\mathbf{b}^t$, and attributes $\mathbf{a}^t$.
9:     $(\mathbf{x}^t, \mathbf{p}^t) \leftarrow \text{MYERSON}(\widetilde{\mathbf{x}}(\cdot), \widetilde{\mathbf{p}}(\cdot))$;
10: **end for**

---

Specifically, the parameter $\nu$ is carefully calibrated to ensure that the bid distribution fed into the private Myerson mechanism is stochastically dominated by the empirical distribution. Our algorithm provides an incentive guarantee that bids lie within a small, controllable neighborhood of the true values. The range of this neighborhood is determined by the privacy parameter $\epsilon_p$ (hence is controlled by our algorithm), and the bidders' utility functions. By setting $\nu$ to match the range of this neighborhood, the resulting distribution is dominated by the empirical distribution.

### 5.3 Revenue Guarantee of the Algorithm

Before presenting the revenue guarantee of our main algorithm, we first introduce a lemma that upper bounds how a bidder's bid deviates from its true value during the collection stage. Intuitively, by the design of our commitment algorithm the bidder will incurs a loss that scales (positively) with the bid deviation, compared to truthful bidding. Furthermore, our private Myerson ensures that the bidder's future utility gain is upper bounded (Lem. J.5). Thus, bidders are incentivized to report bids within a certain range of their true values to optimize their overall utility. More details in Appendix J.4.

**Lemma 5.3** (Bid Deviation). *For any $t \in [0, T_1]$, the bidder will bid only $b_t$ such that $|b_t - v_t| \leq 2\alpha$, where $\alpha = \sqrt{2(l-1)\epsilon_p}hk$ for bidders in a* large market*; and $\alpha = \sqrt{\frac{2\gamma\epsilon_p}{1-\gamma}}kh$ for discounting bidder.*

From this lemma, we get that selecting a small $\epsilon_p$ would incentivize bid distributions that are close to the ground-truth. Let $\nu = 2\alpha$ in our algorithm (line 4, Alg. 3) would yield a distribution that is stochastically dominated by, yet close in revenue guarantee to, the true distribution. Run our DP Myerson algorithm on this distribution would give us sublinear regret, as stated below.

**Theorem 5.4** (Accuracy Guarantee of Two-stage Mechanism). *Given $\epsilon \in [0, 1/4]$, $n$ samples of the joint distribution $\mathbf{D} \in [0, h]^k$, and $T_1 = \Theta(\epsilon^{-2}\log(k/\delta)), T = \Omega(T_1), \epsilon_a = \epsilon_q = \epsilon_p = \epsilon$, with probability $1 - \delta$, Alg. 3 generates sublinear regret, i.e.,*

*Under a* large market*, the regret is upper bounded by $\widetilde{O}[(\epsilon + \sqrt{l\epsilon})kh]$, for $\nu = 2\sqrt{2(l-1)\epsilon_p}hk$.*

*Under discounting bidder, the regret is upper bounded by $\widetilde{O}[(\epsilon + \sqrt{\frac{\gamma\epsilon}{1-\gamma}})kh]$, for $\nu = 2\sqrt{\frac{2\gamma\epsilon_p}{1-\gamma}}kh$.*

*Proof Sketch.* We denote the empirical distribution as $\widehat{D}$, the distribution after subtraction in line 4 of Alg. 3 as $\widetilde{D}$, and the (final) output distribution as $\widehat{D}^p$. Then these distribution satisfies $\widehat{D} \succeq \widetilde{D} \succeq \widehat{D}^p$. By strong monotonicity(Lem. F.3), we know that $\mathbb{E}[\text{Rev}(M_{\widehat{D}^p}, \mathbf{D})] \geq \mathbb{E}[\text{OPT}(\widehat{D}^p)]$. Since $M_{\widehat{D}^p}$ need not be optimal over $\mathbf{D}$, we have that:

$$0 \geq \mathbb{E}[\text{Rev}(M_{\widehat{D}^p}, \mathbf{D}) - \text{OPT}(\mathbf{D})]$$
$$\geq \mathbb{E}[\text{Rev}(M_{\widehat{D}^p}, \mathbf{D}) - \text{OPT}(\widehat{D}^p)] + \mathbb{E}[\text{OPT}(\widehat{D}^p) - \text{Rev}(M_{\mathbf{D}}, \mathbf{D})]$$
$$\geq \mathbb{E}[\text{OPT}(\widehat{D}^p) - \text{OPT}(\widehat{D})] - |\mathbb{E}[\text{OPT}(\widehat{D}) - \text{OPT}(\mathbf{D})]| \geq -\widetilde{\Theta}((\epsilon + \epsilon^2/\epsilon_p)kh + \nu).$$

where in the last inequality we apply revenue shift theorem (Thm. 3.1) to upper bound the first term and apply Lemma J.9 to upper bound the second term. Please refer to Appendix J.3 for more details.

$\square$

## 6 EXPERIMENTS

In this section, we present the experimental results for the Differentially Private (DP) Myerson mechanism, comparing its performance against two standard mechanism design baselines: the *Myerson* (optimal) auction and the *Vickrey* (second-price) auction. The former is designed to achieve near-optimal revenue for a given value distribution, whereas the latter, while strategy-proof, offers no revenue guarantees in settings with independent and non identical value distributions.

Our experiments are conducted on normal and lognormal distributions truncated to positive domains. For each value profile, we test various hyperparameters—additive discretization ($\epsilon_a$), quantile discretization ($\epsilon_q$), and the privacy parameter ($\epsilon_p$)—and select the configuration with the *best* performance. For details on DP Myerson's sensitivity to hyperparameters, see Appendix A.

| Bidder Profile | DP Myerson | Second Price | Myerson | Ref. |
|---|---|---|---|---|
| Normal $\mathcal{N}(0.3, 0.5)$ 
 Lognormal $(\mu, \sigma) = (-1.87, 1.15)$ | 0.25272 | 0.15154 (66.7 %) | 0.32598 | Table 2 |
| Normal $\mathcal{N}(0.3, 0.5)$ 
 Normal $\mathcal{N}(0.5, 0.7)$ | 0.37691 | 0.33741 (11.7 %) | 0.50204 | Table 3 |
| Lognormal $(\mu, \sigma) = (-1.87, 1.15)$ 
 Lognormal $(\mu, \sigma) = (-1.24, 1.04)$ | 0.13912 | 0.11578 (20.2 %) | 0.21292 | Table 4 |

Table 1: Empirical Revenue of DPMyerson (Alg. 1) under 2-dimensional non-identical value distributions. Each DPMyerson configuration is averaged over 50 draws, with revenue evaluated on $10,000$ samples. Percentages in parentheses represent the improvement over the second-price mechanism.

In Table 1, under non i.i.d distribution settings where there is a significant revenue gap between the Vickrey auction and the Myerson auction, DPMyerson achieves a notable revenue increase (at least 11% ) over the second-price mechanism.

## 7 CONCLUSION

We investigate the problem of learning a single-item auction (i.e., Myerson) from samples with *pure* DP. We consider the broader setting where the agents' valuations are *independent*, *non-identical*, and can either be *bounded* or *unbounded*. By recognizing that the optimal auction mechanism exhibits robustness to small statistical perturbations in the underlying distribution, we reduce the challenge of privately learning an optimal auction from sample data to the task of privately approximating pre-specified quantiles. Specifically, our approach ensures pure privacy while generating a distribution that is closely aligned with the underlying distribution in terms of expected revenue.

We then extend this framework to the online auction setting, where later auctions are fitted on bids from previous auctions. In this setting, non-myopic bidders reason about their utility across rounds, and can bid strategically under (one-shot) truthful auctions. By leveraging our private Myerson mechanisms with an extra commitment mechanism, we achieve near-optimal revenue outcomes over the bidders' (unobservable) value samples, despite the strategic complexity introduced by non-myopic behavior. This result highlights the robustness of our approach in both protecting privacy and maintaining near optimal expected revenue in dynamic, strategic environments.

Several interesting directions remain in this line of work.

**Private Correlation Robust Auction Design**. Correlation robust auctions apply to the partial information setting where the distributions are correlated but only the marginal distributions of them can be observed (He and Li, 2022; Bei et al., 2019). Designing a correlation-robust auction with privacy guarantees based on samples would be a valuable future direction.

**Other Bidders' Utility Models**. Other bidder utility functions are also widely considered in the context of online auctions and mechanism design, including the budget-constrained value maximizing bidders (Conitzer et al., 2022), and submodular utility functions (Assadi and Singla, 2020). Advancing privacy-preserving mechanisms for these settings would also be important.

ACKNOWLEDGEMENTS

We sincerely thank the anonymous reviewers from ICLR 2025 for their valuable discussions and suggestions on our experiments, which have improved this work. Yuanyuan Yang and Jamie Morgenstern are supported by NSF Awards CCF-2045402 and CCF-2019844.

We gratefully acknowledge Bhuvesh Kumar for his contributions to the early discussions of this work prior to 2022, at a time when privacy techniques had not yet been developed. However, Bhuvesh Kumar was not involved in the substantial revisions and restructuring of the manuscript that took place from 2022 onward.

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
