non-i.i.d. value distributions. All reported results are based on experiments conducted with at least $10,000$ samples. In the following subsections, we detail the various experimental settings and configurations considered in this study.

### A.1  DISTRIBUTION

**Definition A.1** (Log Normal Distribution). A random variable $V$ is said to be lognormal distributed with parameter $(\mu, \sigma)$, if $\ln(V)$ follows normal distribution $\mathcal{N}(\mu, \sigma)$.

Given mean $\mu_V$ and and standard deviation $\sigma_V$, the parameter of the underlying normal distribution is as follows:

$$\mu = \ln\left(\frac{\mu_V^2}{\sqrt{\mu_V^2 + \sigma_V^2}}\right), \sigma = \sqrt{\ln\left(1 + \frac{\sigma_V^2}{\mu_V^2}\right)}$$

### A.2  EMPIRICAL REVENUE ANALYSIS OF DP MYERSON ACROSS HYPERPARAMETERS

In this subsection, we evaluate the performance of DP Myerson across various hyperparameter configurations, including additive discretization ($\epsilon_a$), privacy parameter ($\epsilon_p$), and quantile discretization ($\epsilon_q$), using 2-dimensional non-i.i.d. distributions. We compare the revenue guarantees of DP Myerson against those of the second-price auction and the Myerson auction.

| Bidder 1 | Normal $\mathcal{N}(0.3, 0.5)$ | | | |
|---|---|---|---|---|
| Bidder 2 | Lognormal $(\mu, \sigma) = (-1.87, 1.15)$ | | | |
| Privacy Parameter $\epsilon_p$ | 0.1 | 0.2 | 0.4 | 0.7 |
| | | | | |
| *w/ additive $\epsilon_a = 0.05$, Upper Bound $h = 1$.* | | | | |
| DPMy w/ $\epsilon_q = 0.05$ | 0.14974 | 0.15777 | 0.17104 | 0.14598 |
| DPMy w/ $\epsilon_q = 0.1$ | 0.15564 | 0.17322 | 0.17218 | **0.18670** |
| DPMy w/ $\epsilon_q = 0.14$ | **0.18674** | **0.19108** | **0.18960** | 0.18429 |
| Second Price | 0.17755 | | | |
| Myerson | 0.33291 | | | |
| *w/ additive $\epsilon_a = 0.1$, Upper Bound $h = 1$.* | | | | |
| DPMy w/ $\epsilon_q = 0.26$ | 0.2500 | **0.25272** | **0.248056** | 0.24658 |
| DPMy w/ $\epsilon_q = 0.31$ | **0.24780** | 0.24387 | 0.24540 | 0.24689 |
| DPMy w/ $\epsilon_q = 0.36$ | 0.24534 | 0.24899 | 0.24687 | **0.24723** |
| Second Price | 0.15154 | | | |
| Myerson | 0.32598 | | | |

Table 2: Average Empirical Revenue of DP Myerson under non-i.i.d. Value Distributions with Varying Discretization Parameters (additive $\epsilon_a$, quantile $\epsilon_q$) and Privacy Parameter $\epsilon_p$. The performance of each DP Myerson is averaged over 50 draws, fitted on $100,000$ samples, with empirical average revenue evaluated over another $10,000$ samples. Best performing for each $\epsilon_p$ are marked **bold**.

In Table 2, we evaluate the performance of DPMyerson in a setting where one bidder's value distribution follows a normal distribution, and the other follows a lognormal distribution with parameters $(\mu, \sigma) = (-1.87, 1.15)$ (i.e., mean 0.3 and standard deviation 0.5). The revenue achieved by the best DPMyerson configuration significantly exceeds that of the second-price auction.

| Bidder 1 | Normal $\mathcal{N}(0.3, 0.5)$ | | | |
| Bidder 2 | Normal $\mathcal{N}(0.5, 0.7)$ | | | |
| Privacy Parameter $\epsilon_p$ | 0.1 | 0.2 | 0.4 | 0.8 |
| | | | | |
| *w/ additive $\epsilon_a = 0.1$, Upper Bound $h = 1.5$, w/ $100,000$ samples.* | | | | |
| DPMy w/ $\epsilon_q = 0.05$ | 0.33918 | 0.32878 | 0.34927 | 0.33501 |
| DPMy w/ $\epsilon_q = 0.2$ | 0.36784 | 0.36229 | 0.36408 | **0.37160** |
| DPMy w/ $\epsilon_q = 0.3$ | **0.37521** | **0.37691** | **0.37436** | 0.35579 |
| Second Price | 0.33741 | | | |
| Myerson | 0.50204 | | | |

Table 3: Average Empirical Revenue of DP Myerson under non-i.i.d. Value Distributions with Varying Discretization Parameters (additive $\epsilon_a$, quantile $\epsilon_q$) and Privacy Parameter $\epsilon_p$. The performance of each DP Myerson is averaged over 50 draws, with empirical average revenue evaluated over another $10,000$ samples. Best performing for each $\epsilon_p$ are marked **bold**.

In Table 3, we evaluate the performance of DPMyerson under non-i.i.d. normal bid distributions. The best DPMyerson configuration achieves a significant revenue improvement over the second-price auction, with an increase of 12%.

| Bidder 1 | Lognormal $(\mu, \sigma) = (-1.8685, 1.1528)$ | | | |
| Bidder 2 | Lognormal $(\mu, \sigma) = (-1.2357, 1.0417)$ | | | |
| Privacy Parameter $\epsilon_p$ | 0.1 | 0.2 | 0.4 | 0.8 |
| | | | | |
| *w/ additive $\epsilon_a = 0.1$, Upper Bound $h = 1.0$.* | | | | |
| DPMy w/ $\epsilon_q = 0.1$ | 0.13536 | **0.13156** | 0.11881 | 0.11976 |
| DPMy w/ $\epsilon_q = 0.2$ | **0.13912** | 0.13448 | **0.12737** | **0.13947** |
| DPMy w/ $\epsilon_q = 0.3$ | 0.035531 | 0.03761 | 0.03568 | 0.03952 |
| Second Price | 0.11578 | | | |
| Myerson | 0.21292 | | | |

Table 4: Average Empirical Revenue of DP Myerson under non-i.i.d. Value Distributions with Varying Discretization Parameters (additive $\epsilon_a$, quantile $\epsilon_q$) and Privacy Parameter $\epsilon_p$. The performance of each DP Myerson is averaged over 50 draws, fitted on $100,000$ samples, with empirical average revenue evaluated over another $10,000$ samples. Best performing for each $\epsilon_p$ are marked **bold**.

In Table 4, we evaluate the performance of DPMyerson under non-i.i.d. lognormal bid distributions. The first bidder's value distribution follows a lognormal distribution with parameters $(\mu, \sigma) = (-1.8685, 1.1528)$ (mean 0.3, std 0.5), while the second bidder's value distribution follows a lognormal distribution with parameters $(\mu, \sigma) = (-1.2357, 1.0417)$ (mean 0.5, std 0.7).

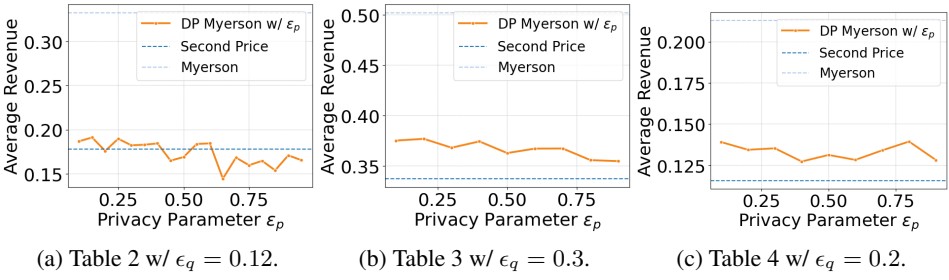

(a) Table 2 w/ $\epsilon_q = 0.12$.     (b) Table 3 w/ $\epsilon_q = 0.3$.     (c) Table 4 w/ $\epsilon_q = 0.2$.

Figure 3: The impact of different privacy parameter $\epsilon_p$ w/ other discretization parameters fixed.

In Figure 3, we illustrate the revenue performance of DP Myerson across different values of the privacy parameter $\epsilon_p$. As shown, $\epsilon_p$ acts as a hyperparameter, and the revenue does not vary monotonically with changes in $\epsilon_p$. Interestingly, while one might intuitively expect higher values of $\epsilon_p$ (indicating weaker privacy constraints) to result in consistently improved revenue, this is not observed uniformly. Instead, the relationship between $\epsilon_p$ and revenue appears non-linear, suggesting a complex interplay between privacy guarantees and auction performance.

Notably, the performance of DP Myerson demonstrates significant robustness to the choice of $\epsilon_p$ under *large* sample sizes. This robustness implies that careful tuning of $\epsilon_p$ may not always be critical for achieving competitive revenue, particularly in data-rich settings. However, for smaller sample sizes, the choice of $\epsilon_p$ could have a more pronounced impact, potentially requiring more nuanced calibration to balance privacy and revenue effectively.

### A.3    EMPIRICAL REVENUE ANALYSIS OF DP MYERSON WITH ADDED BENCHMARKS

In this subsection, we evaluate the performance of DP Myerson across additional hyperparameter configurations, with added benchmarks. The revenue of Myerson and Second Price auctions are evaluated over the same distribution as DP Myerson, which are additively discretized by $\epsilon_a$ and truncated by $h$[6]. These parameters are specified per table.

As we introduced non-conventional *quantile discretization*, our experiments in this section will focus more on the loss under QE and DPQE, rather than the analysis of the revenue loss by additive discretization. Consequently, we added two benchmarks: 1) The first one is the theoretical lower bound for DP Myerson, calculated by the revenue of empirical Myerson under additively discretized distribution minus our revenue loss upper bound of quantile discretization $\epsilon_q h$. 2) The second one is the Myerson under the same quantile and additive discretization, but without the privacy estimation step. The first benchmark assesses whether our experimental results align with theoretical expectations. The second benchmark evaluates the accuracy loss introduced by the DP step.

| Bidder 1 | | Uniform $\mathcal{U}(0, 0.4)$ | | |
|---|---|---|---|---|
| Bidder 2 | | Uniform $\mathcal{U}(0, 0.6)$ | | |
| Quantile Discretization $\epsilon_q$ | 0.05 | 0.15 | 0.25 | 0.3 |
| | | | | |
| *w/ additive* $\epsilon_a = 0.05$, Upper Bound $h = 0.6$, w/ $100,000$ samples. | | | | |
| DPMy w/ $\epsilon_p = 0.1$ | 0.20322 | 0.18411 | 0.20093 | **0.207881** |
| DPMy w/ $\epsilon_p = 0.2$ | 0.20262 | 0.180435 | **0.20357** | 0.199048 |
| DPMy w/ $\epsilon_p = 0.3$ | **0.20545** | 0.187785 | 0.19146 | 0.17676 |
| Theoretical LB for DPMy | 0.17411 | 0.11411 | 0.05411 | 0.02411 |
| Myerson w/ QD | 0.20435 | **0.19973** | 0.20285 | 0.19448 |
| Second Price | | 0.13238 | | |
| Myerson | | 0.20411 | | |

Table 5: Average Empirical Revenue of DP Myerson under non-i.i.d. Value Distributions with Varying Quantile Discretization Parameters $\epsilon_q$ and Privacy Parameter $\epsilon_p$. The performance of each DP Myerson is averaged over 5 draws, with empirical average revenue evaluated over another $10,000$ samples. The best-performing value for each $\epsilon_q$ is marked in **bold**, and the best-performing DPMyerson (if different) for each $\epsilon_q$ is underlined.

In Table 5, we analyze the performance of DP-Myerson under non-i.i.d. uniform bid distributions. The best DP-Myerson configuration here achieves a significant revenue improvement over the second-price auction, with an increase of 56%. In addition, the performance of the optimal DP-Myerson (w.r.t the best privacy parameter) does not vary significantly with the choice of quantile discretization. We believe this is because the privacy parameter acts as a regularizer that prevents overfitting.

---

[6]The reason for applying the Myerson under the discretized distribution is that for continuous distribution, the pseudo-dimension of the is infinity, and hence is impossible to learn without discretization.

| Bidder 1 | Normal $\mathcal{N}(0.2, 0.15)$ | | |
| Bidder 2 | Normal $\mathcal{N}(0.25, 0.1)$ | | |
| Quantile Discretization $\epsilon_q$ | 0.1 | 0.12 | 0.18 |
| | | | |
| *w/ additive $\epsilon_a = 0.05$, Upper Bound $h = 0.5$, w/ 100,000 samples.* | | | |
| DPMy w/ $\epsilon_p = 0.1$ | **0.157098** | 0.173557 | 0.185344 |
| DPMy w/ $\epsilon_p = 0.2$ | 0.131482 | 0.174008 | 0.140837 |
| DPMy w/ $\epsilon_p = 0.3$ | 0.141554 | 0.163951 | 0.149813 |
| Theoretical LB for DPMy | 0.13788 | 0.12788 | 0.09788 |
| Myerson w/ QD | 0.14752 | **0.18641** | **0.186455** |
| Second Price | | 0.14960 | |
| Myerson | | 0.18788 | |

Table 6: Average Empirical Revenue of DP Myerson under non-i.i.d. Value Distributions with Varying Quantile Discretization Parameters $\epsilon_q$ and Privacy Parameter $\epsilon_p$. The performance of each DP Myerson is averaged over 5 draws, with empirical average revenue evaluated over another 10,000 samples. The best-performing value for each $\epsilon_q$ is marked in **bold**, and the best-performing DPMyerson (if different) for each $\epsilon_q$ is underlined.

In Table 6, we analyze the performance of DP-Myerson under non-i.i.d. normal bid distributions. The best DP-Myerson configuration here achieves a significant revenue improvement over the second-price auction, with an increase of 25%.

| Bidder 1 | Normal $\mathcal{N}(0.2, 0.15)$ | | |
| Bidder 2 | Lognormal $(\mu, \sigma) = (-2.08, 0.606)$ | | |
| Quantile Discretization $\epsilon_q$ | 0.05 | 0.2 | 0.25 |
| | | | |
| *w/ additive $\epsilon_a = 0.05$, Upper Bound $h = 0.5$, w/ 100,000 samples.* | | | |
| DPMy w/ $\epsilon_p = 0.1$ | 0.10910 | 0.10889 | **0.12175** |
| DPMy w/ $\epsilon_p = 0.2$ | 0.11116 | 0.11380 | 0.11843 |
| DPMy w/ $\epsilon_p = 0.4$ | 0.08487 | 0.12480 | 0.11990 |
| Theoretical LB for DPMy | 0.11535 | 0.04035 | 0.01535 |
| Myerson w/ QD | **0.13692** | 0.09755 | 0.11479 |
| Second Price | | 0.09494 | |
| Myerson | | 0.14035 | |

Table 7: Average Empirical Revenue of DP Myerson under non-i.i.d. Value Distributions with Varying Quantile Discretization Parameters $\epsilon_q$ and Privacy Parameter $\epsilon_p$. The performance of each DP Myerson is averaged over 5 draws, with empirical average revenue evaluated over another 10,000 samples. The best-performing value for each $\epsilon_q$ is marked in **bold**, and the best-performing DPMyerson (if different) for each $\epsilon_q$ is underlined.

In Table 7, we analyze the performance of DP-Myerson under non-i.i.d. lognormal bid distributions with $(\mu, \sigma) = (-2.08, 0.606)$. These parameters also indicates that this lognormal distribution has mean 0.15 and standard deviation 0.1. The best DP-Myerson configuration here achieves a significant revenue improvement over the second-price auction, with an increase of 44%.

In Table 8, we analyze the performance of DP-Myerson under non-i.i.d. lognormal and normal bid distributions with $(\mu, \sigma) = (-1.87, 1.15)$. These parameters also indicates that this lognormal distribution has mean 0.3 and standard deviation 0.5. The best DP-Myerson configuration here achieves a significant revenue improvement over the second-price auction, with an increase of 28%. Specifically, for particular configurations, our theoretical lower bound is actually 0. Therefore, the theoretical lower bound of quantile estimation is significantly lower compared to what we observe in experiments.

| Bidder 1 | Normal $\mathcal{N}(0.3, 0.5)$ | | |
| Bidder 2 | Lognormal $(\mu, \sigma) = (-1.87, 1.15)$ | | |
| Quantile Discretization $\epsilon_q$ | 0.25 | 0.35 | 0.45 |
| | | | |
| *w/ additive $\epsilon_a = 0.1$, Upper Bound $h = 1$, w/ $100,000$ samples.* | | | |
| DPMy w/ $\epsilon_p = 0.1$ | **0.22681** | **0.23123** | **0.23517** |
| DPMy w/ $\epsilon_p = 0.2$ | 0.21735 | 0.22713 | 0.2264 |
| DPMy w/ $\epsilon_p = 0.3$ | 0.196502 | 0.22713 | 0.22924 |
| Theoretical LB for DPMy | 0.0332 | 0 | 0 |
| Myerson w/ QD | 0.20407 | 0.21472 | 0.12004 |
| Second Price | 0.14211 | | |
| Myerson | 0.28323 | | |

Table 8: Average Empirical Revenue of DP Myerson under non-i.i.d. Value Distributions with Varying Discretization Parameters (additive $\epsilon_a$, quantile $\epsilon_q$) and Privacy Parameter $\epsilon_p$. The performance of each DP Myerson is averaged over 5 draws, with empirical average revenue evaluated over another $10,000$ samples. The best-performing value for each $\epsilon_q$ is marked in **bold**, and the best-performing DPMyerson (if different) for each $\epsilon_q$ is underlined.

In Table 9, we analyze the performance of DP-Myerson under non-i.i.d. lognormal bid distributions with $(\mu, \sigma) = (-1.2357, 1.0417)$, and $(\mu, \sigma) = (-1.87, 1.15)$. The best DP-Myerson configuration here achieves a significant revenue improvement over the second-price auction, with an increase of 49%. Specifically, our theoretical lower bound is 0 for *all* configurations, indicating that practical revenue loss of private quantile estimation is much smaller than its theoretical lower bound.

In Table 10, we analyze the performance of DP-Myerson under non-i.i.d. normal bid distributions. The best DP-Myerson configuration here achieves a significant revenue improvement over the second-price auction, with an increase of 21%. Specifically, our theoretical lower bound is 0 for *all* configurations, implying that the practical revenue loss in private quantile estimation is far smaller than the theoretical bound suggests.

| Bidder 1 | Lognormal $(\mu, \sigma) = (-1.2357, 1.0417)$ | | |
| Bidder 2 | Lognormal $(\mu, \sigma) = (-1.87, 1.15)$ | | |
| Quantile Discretization $\epsilon_q$ | 0.25 | 0.3 | 0.35 |
| | | | |
| *w/ additive $\epsilon_a = 0.1$, Upper Bound $h = 1$, w/ $100,000$ samples.* | | | |
| DPMy w/ $\epsilon_p = 0.1$ | 0.15682 | 0.15232 | 0.15528 |
| DPMy w/ $\epsilon_p = 0.2$ | **0.17156** | 0.15266 | **0.16803** |
| DPMy w/ $\epsilon_p = 0.3$ | 0.16144 | **0.16551** | 0.15901 |
| Theoretical LB for DPMy | 0 | 0 | 0 |
| Myerson w/ QD | 0.0843 | 0.14279 | 0.09735 |
| Second Price | 0.11545 | | |
| Myerson | 0.2165 | | |

Table 9: Average Empirical Revenue of DP Myerson under non-i.i.d. Value Distributions with Varying Quantile Discretization Parameters $\epsilon_q$ and Privacy Parameter $\epsilon_p$. The performance of each DP Myerson is averaged over 5 draws, with empirical average revenue evaluated over another $10,000$ samples. The best-performing value for each $\epsilon_q$ is marked in **bold**, and the best-performing DPMyerson (if different) for each $\epsilon_q$ is underlined.

From these experiments with added benchmarks, we concluded with the following takeaways:

- The empirical revenue of *all* DPMyerson and Myerson with Quantile discretization (QD) is above the theoretical lower bounds for the Myerson on quantile discretized distributions.

| | | | |
|---|---|---|---|
| Bidder 1 | Normal $\mathcal{N}(0.3, 0.5)$ | | |
| Bidder 2 | Lognormal $(\mu, \sigma) = (-1.87, 1.15)$ | | |
| Quantile Discretization $\epsilon_q$ | 0.35 | 0.4 | 0.45 |
| | | | |
| *w/ additive* $\epsilon_a = 0.1$, Upper Bound $h = 1$, w/ $100,000$ samples. | | | |
| DPMy w/ $\epsilon_p = 0.1$ | **0.26716** | **0.26900** | **0.26995** |
| DPMy w/ $\epsilon_p = 0.2$ | 0.25675 | 0.26339 | 0.26339 |
| DPMy w/ $\epsilon_p = 0.3$ | 0.26707 | 0.26622 | 0.25957 |
| Myerson w/ QD | 0.21953 | 0.24543 | 0.16873 |
| Theoretical LB for DPMy | 0 | 0 | 0 |
| Second Price | 0.22132 | | |
| Myerson | 0.33482 | | |

Table 10: Average Empirical Revenue of DP Myerson under non-i.i.d. Value Distributions with Varying Quantile Discretization Parameters $\epsilon_q$ and Privacy Parameter $\epsilon_p$. The performance of each DP Myerson is averaged over 5 draws, with empirical average revenue evaluated over another $10,000$ samples. The best-performing value for each $\epsilon_q$ is marked in **bold**, and the best-performing DPMyerson (if different) for each $\epsilon_q$ is underlined.

- We notice that for value profiles where the gap between DP Myerson and Myerson are large, the gap between Myerson and Myerson w/ QD are also large. In addition, this loss is *inherent* to the properties of the distribution, and will become larger if the upper bound is considerably large compared to the optimal revenue from the same distribution.

- The performance of DP Myerson obtains similar or superior performance compared to the Myerson under the same quantile discretization, meaning that the revenue loss due to privacy doesn't contribute too much to the overall revenue loss.

- The revenue loss of Myerson's with QD (and DP Myerson) doesn't always decrease w/ the increase of quantile discretization, and can exceed the revenue by Myerson evaluated on fresh samples of the same distribution. We conjectured that this is because here the QD and privacy both serves as regularizers that prevents from overfitting.

## B    REMARKS

### B.1    GENERALIATION TO JOINT DIFFERENTIAL PRIVACY

A related but weaker privacy notion for *multi-player* setting, i.e, jointly differential privacy (JDP) (Kearns et al., 2014) also applies to our setting. Standard Differential Privacy (DP) requires that changing one entry in the dataset affects the probability of every possible output *vector* by at most $\epsilon_p$. In contrast, joint differential privacy only requires that changes in one player's input (multi-party collision) do not significantly affect other player's *scalar* outcome, without imposing restrictions on how those changes impact the outcomes of the players whose inputs were altered. Thus, DP implies JDP, and our algorithm provides a JDP guarantee that is $1/k$ of its DP guarantee.

Notice that JDP doesn't implies DP in general. JDP guarantees privacy only in an incomplete information setting where each bidder sees only their own outcome, which is often impractical in auction settings due to (1) Sybil attacks, where bidders may create multiple identities/attributes within the same auction, and (2) transparency requirements, such as EU regulations mandating public disclosure of political ad payments and allocations (European Commission, 2024).

### B.2    GENERALIZATION OF DP MYERSON

Our DP Myerson for unbounded distributions (Alg. 9) can be generalized to unbounded and irregular distribution settings with *light tail*. The effectiveness of the truncation depends solely on whether the tail of the distribution decays faster than that of the equal revenue distribution, which has support over $[1, +\infty]$ and has a cdf $F(v) = 1 - 1/v$. When the condition is met, the truncated distribution (hence the integrated algorithm), still approximately maintains the revenue guarantee.

### B.3    GENERALIZATION TO OTHER ONLINE AUCTION SETTINGS

**Varying Bidder Counts**. Our mechanism generalizes to settings where, in certain iterations, there are multiple bidders or occurrences for a single attribute or an absence of a single attribute, as long as there are sufficient samples for each attribute over the collection stage.

**Generalization to Bounded Rationality**. If we adopt the weaker assumption that the bidders has bounded rational, i.e., they will bid truthfully if strategic bidding only gives them at most a small fraction of the extra utility. Then running our DP Myerson *alone* with an appropriate privacy parameter $\epsilon_p$ would incentivize truthful bidding. In this context, the commitment mechanism can be replaced by any truthful and prior-independent auctions, for example, second price auction.

## C   PRIOR WORK DISCUSSION

**DP Mechanism Design.**   Emerging from McSherry and Talwar (2007), there has been interest in delivering mechanisms with DP guarantees Nissim et al. (2012). However, their designs focused more on approximating optimal utility and less on running time efficiency. Consequently, these algorithms incur exponential time in our setting, even when the distribution is of finite support(See Appendix D).

The most relevant recent work is  Huang et al. (2018a), which developed an $(\epsilon, \delta)$-approximate private empirical reserve mechanism by applying the Gaussian mechanism via two-fold aggregation (Dwork et al., 2010). The added noise follows a mean-0 normal distribution and hence coincides with the smoothed analysis framework, allowing this work to apply solutions from there (Abernethy et al., 2014). However, getting a stronger pure DP mechanism requires the added noise to be *non-normal*. Thus, existing technical solutions from smoothed analysis do not apply to our setting.

**Sample Complexity of Auctions**   One line of related research problem is to show provably sample complexity guarantees for learning in auctions from *truthful* samples, i.e., how many samples are needed to learn auctions that approximately maximize revenue. This problem was first introduced by (Balcan et al., 2005), and led to an explosion of work on the topic (Bubeck et al., 2019; Huang et al., 2018b; Morgenstern and Roughgarden, 2015; Elkind, 2007; Balcan et al., 2005; 2008; Roughgarden and Schrijvers, 2016; Cai and Daskalakis, 2011; Hartline and Roughgarden, 2009; Devanur et al., 2016; Balcan et al., 2016; Cole and Roughgarden, 2014; Guo et al., 2019). Typically, they upper bound the sample complexity of certain auctions by proposing a mechanism that achieves the proposed complexity, and their lower bound for independent, non-i.i.d single-item auctions apply to our setting.

However, it is non-trivial to extend their mechanism to our setting with non-myopic bidders, in that this line of work assumes the learner/ auctioneer has access to *truthful samples*. In contrast, in our setting, if the bidders participate in multiple rounds of the auction, they can bid strategically to maximize their own total utility over all their rounds, and hence the samples are no longer truthful.

**Online learning in repeated auction**   Reserve-price style strategies achieve near-optimal revenue for the i.i.d setting and can be learned within given incentive guarantees (Deng et al., 2020; Kanoria and Nazerzadeh, 2021). However, these methods capture only a constant fraction of the optimal revenue in our setting where the bidders are from *different* distribution.[7]

Huang et al. (2018a); Abernethy et al. (2019) have applied differential privacy as a solution to achieve incentive compatibility. However, their methods rely heavily on the existence of an upper bound of the value distribution and are thus not applicable to the unbounded setting.

**Incentive measurements**   Another relevant topic is to measure and guarantee the incentive compatibility (IC) Myerson (1979) in a mechanism. A mechanism is IC if truthful bidding outperforms other strategies. In the absence of truthful samples from value distribution, strict-IC and the optimality of revenue guarantee cannot be achieved simultaneously.

Hence, previous works have designed several approximate IC metrices and methods to evaluate them from samples (Balcan et al., 2019). This includes approximate Bayesian Incentive Compatibility Balseiro et al. (2024), approximate Dominant Strategy Incentive Compatibility Dütting et al. (2024), Stage Incentive Compatibility Deng et al. (2021), and etc.

**Quantile Estimation**   Our paper also incorporates the use of quantile estimation oracles, a problem relevant in both offline and streaming settings (Zhang et al., 2006; Gupta et al., 2024; Gribelyuk et al., 2025). Extensive research has been conducted on quantile estimation with differential privacy guarantees (Smith, 2011; Duchi et al., 2018; Gillenwater et al., 2021; Alabi et al., 2022; Lalanne et al., 2023; Liu et al., 2023; Tran et al., 2024), and our work builds upon the methodologies of Durfee (2023) and Kaplan et al. (2022). Notably, their guarantees transfer to the statistical Kolmogorov-Smirnov (K-S) distance when the measured data points follow the same distribution.

---

[7]See example 3.11 in Hartline and Roughgarden (2009).

## D  FAILED ATTEMPTS

### D.1  FAILED ATTEMPTS FOR DP MYERSON

**Failed Attempt 1: Deploying the Exponential Algorithm**. Incorporating DP into a mechanism often reduces efficiency, as privacy guarantees introduce computational overhead (e.g., noise addition or extra sampling). Similar trade-offs arise in online learning (Jain et al., 2012), federated learning (Zhang et al., 2023), and deep learning (Abadi et al., 2016).

The exponential mechanism (McSherry and Talwar, 2007) is one typical solution to integrate pure DP with Myerson, but applying it directly to a continuous distribution is computationally inefficient. This requires fitting an (ironed) virtual value curve for each dimension—a continuous function with an unknown exact form, except that it is monotonically increasing with value.

A plausible fix is to discretize the distribution into a finite number of values, and then deploy exponential over possible Myerson's over the discretized distribution, assuming the distribution is bounded. This doesn't fully resolve the computational challenges. Specifically, when each value distribution is of finite support $l$ and there are $k$ different attributes/distributions, then Myerson's auction corresponds to ranking all $kl$ values in increasing order of their virtual values. According to this ranking, the mechanism picks the bidder with the highest-ranked value as the winner and charges them the min value that maintains a higher rank than the value of the second-highest-ranked bid.

Thus, the exponential mechanism corresponds to sampling an ordering over all $(kl)!/(l!)^k$ possible rankings, with each requiring $O(kn)$ time to evaluate revenue, resulting in *exponential* running time. Suppose we additive discretize each (bounded) distribution by $\epsilon_a$, the number of distinct ordering will be $\Theta(k^{k/\epsilon_a})$, which blow up exponentially with $1/\epsilon_a$). This approach is inefficient since the number of possible rankings grows exponentially with $l$ and does not generalize to unbounded distributions.

**Failed Attempt 2: Pre-processing Via Tree Aggregation**. One might wonder whether we could use tree aggregation with differentially private(DP) noise (e.g., Laplacian noise) on its cumulative density function after discretizing the value space into intervals when value distribution is bounded. While this method could maintain DP guarantee, the noise added only maintains "close" approximations to the zeroth order information of the revenue curve, and could lead to *negative probability mass* on certain intervals. This solution indeed is feasible for mechanisms that only require the accuracy of the zeroth order information of the revenue curve, e.g., empirical reserve. However, for the Myerson auction in the non i.i.d case, approximating this mechanism requires both the zeroth order and the first order information of the revenue curve, hence tree aggregation is not feasible.

**Failed Attempt 3: Postprocessing**. Another common solution to differentially private release mechanisms is to add noise to the output. Although this method witnessed its success in robust and differentially private mean estimation, it's unclear how to handle noise in the mechanism design setting even when the value distribution is bounded by, say $H$. The reason is due to sensitivity: For mean estimation, the sensitivity (hence the necessary level of added noise ) grows smaller with a larger number of samples. On the contrary, for the mechanism design setting, it's not clear how the efficiency guarantee (e.g. revenue, social welfare) scales with the inverse of the number of bidders. [8] Unfortunately, if we add DP noise according to sensitivity $H$, the noise level is too large and fails to guarantee a near-optimal target.

### D.2  FAILED ATTEMPTS AND LOWER BOUNDS FOR ONLINE MECHANISM DESIGN

If the bidder is non-discounting and participates in every round of the auction, then it is known that it is not possible to obtain sublinear regret against bid history.

**Lemma D.1** (Regret Lower Bound for Additive Bidder (Theorem 3, Amin et al. (2013))). *Let $\mathcal{A}$ be any seller algorithm for the repeated setting; then, there exists a valuation $\mathcal{D}$ such that:*

$$\text{Regret} \geq 1/12.$$

For application to our online setting, bounding the regret against *bid* history is not enough to guarantee a near-optimal revenue for the value distribution.

---

[8] In fact, this efficiency guarantee in the worst case will be exactly $H$.

**Failed Attempt: Bounding the Regret Only**    Another way of thinking about the repeated auction problem is to reduce it to the online setting, and the benchmark is the revenue produced by best-fixed mechanism over the *bid history*. At every iteration, however, the adversary would produce bids as the best (or better) response to the mechanism, hence this bid could deviate *a lot* from the value sequence. Thus, the revenue from the benchmark could be arbitrarily worse than the revenue from the best-fixed mechanism over the *value history*.

# E    MORE DETAILS FROM PRELIMINARIES

**Notation.**    For a mechanism $M$ and a $k$ dimensional product distribution $\mathbf{D} = D_1 \times \cdots \times D_k$, denote $\mathsf{Rev}(M, \mathbf{D})$ as the expected revenue by running auction $M$ on $\mathbf{D}$. Let $M_{\mathbf{D}}$ be the revenue-maximizing auction on $\mathbf{D}$ and its expected revenue be $\mathsf{OPT}_{\mathbf{D}} = \mathsf{Rev}(M_{\mathbf{D}}, \mathbf{D})$. Denote $x_j(v) \in [0, 1], p_j(v)$ as the allocation probability and the payment for the bidder with value $v$ and distribution $\mathcal{D}_j$. We overload $j$ to denote both the index of bidder and the index of distribution, and we denote $\mathbf{x} = (x_1, \ldots, x_k)$ and $\mathbf{p} = (p_1, \ldots, p_k)$ as the allocation vector and payment vector, respectively. We use $\mathsf{Rev}(M, D)$ to denote the expected revenue for mechanism $M$ on distribution $D$, where $M = (\mathbf{x}, \mathbf{p})$ denotes the allocation and payment as a function of bids.

For bidder $j$ at round $t \in [T]$, denote the bidder value as $v_j^t$, bid as $b_j^t$, the allocation rule as $x_j^t$, and the set of indices of rounds the bidder participates in the mechanism as $S_j$. WLOG, for $t \notin S_j$, we let $b_j^t = v_j^t = 0$. We denote $v_j^{[T]} = (v_j^1, \ldots, v_j^T)^\top$ and $b_j^{[T]} = (b_j^1, \ldots, b_j^T)^\top$ as the batched value and bid vectors, respectively. Assume the utility of bidder $j$ at time $t$ with bid $b_j^t$ is $u_j^t(b_j^t) = x_j^t(b_j^t) \cdot (v_j^t - p_j^t)$ and over *all* $T$ rounds is $u_j^{[T]}(b_j^{[T]}, v_j^{[T]}, h_j^t) = \sum_{t \in S_j} x_j^t(b_j^t) \cdot (v_j^t - p_j^t)$. Denote $U_j^t(b_j^{[T]}) = \mathbb{E}[u_j^{[T]}(b_j^{[T]}, v_j^{[T]}, h_j^t) - u_j^{[t]}(b_j^{[T]}, v_j^{[T]}, h_j^t)]$ as the expected utility of bidder $j$ from the $t$-th round to final round $T$ if the bidder's bid vector is $b_j^{[T]}$, where $h_j^t$ is the history at the $t$-th round, including the price history up to the $t$-th round.

## E.1    MECHANISM DESIGN BASICS

In this section, we present a detailed definitions on the machineries we use in this paper. We begin with the formal definition of Myerson's auction, which maximizes revenue in Bayesian environments.

**Definition E.1** (Myerson's auction, formal version of Definition 2.2)**.** Myerson (1981) Myerson's auction maximizes the expected revenue of a single-item single round auction on product distribution $\mathbf{D} = \mathcal{D}_1 \times \ldots \times \mathcal{D}_k$. Consider the single round auction where there $k$ bidders, where bidder $i$ is from distribution $\mathcal{D}_i$, and let $F_i$ and $f_i$ denote the cdf and pdf of her value distribution.

For continuous product distribution $\mathbf{D}$, the virtual value $\phi_i(v_i)$ of the bidder $i$ with value $v_i$ is $\phi_i(v_i) = v_i - \frac{1 - F_i(v_i)}{f_i(v_i)}$. For the case where the product distribution $\mathbf{D}$ is discrete, the virtual value function from distribution $D_j$ at value $v_i^j$ with support $\mathcal{V}_j = \{v_1^j, \ldots, v_n^j\}$, is defined as(Elkind (2007)):

$$\phi_j(v_i^j, v_{i+1}^j) = v_i^j - (v_{i+1}^j - v_i^j)\frac{1 - F_j(v_i^j)}{f_j(v_i^j)}$$

where $v_i^j$s are ordered in increasing order of $i$, and $f_j(v_i^j) = \mathbb{P}[v^j = v_i^j]$, and $F_j(v_i^j) = \sum_{k=1}^i f(v_k^j)$.

We say a distribution $\mathcal{D}_j$ is $\eta-$strongly regular if for every distribution $j$, $\phi_j(v_i) - \phi_j(v_j) \geq \eta(v_i - v_j)$, for every $v_i > v_j \in \mathcal{V}$. When $\eta = 1$, we say the distribution is monotone hazard rate(MHR) distribution. For any distribution $\mathcal{D}$ with the above property, Myerson's allocation rule is

$$x_i(v_i) = \begin{cases} 1 \text{ if } \phi_i(v_i) \geq \max(0, \max_{j \neq i} \phi_j(v_j)) \\ 0 \text{ otherwise} \end{cases}$$

and payment[9] rule is

$$p_i(v_i) = \begin{cases} \phi_i^{-1}(\max(0, \max_{j \neq i} \phi_j(v_j))) \text{ if } \phi_i(v_i) \geq \max(0, \max_{j \neq i} \phi_j(v_j)). \\ 0 \text{ otherwise} \end{cases}$$

---

[9]The virtual value inverse $\phi_i^{-1}(v)$ for *discrete* distribution is defined as $\arg\min_{v \in \mathcal{V}} \phi_i(v) \geq \phi$, where $\mathcal{V}$ is the support for distribution $\mathcal{D}_i$.

For regular distributions, Myerson's auction allocates the good to the bidder with highest non-negative virtual value[10], and the winner pays the threshold value[11]. If there are no bidders with non-negative virtual value, no one wins the item.

For irregular distributions, Myerson's auction requires an extra "ironing" procedure. The "ironed" virtual value $\widetilde{\phi}$ is a monotonic increasing function of the value, and the above payment/allocation rule are defined based on the "ironed" virtual value.

Next, we introduce the definition of Vickrey auction.

**Definition E.2** (Vickrey Auction). For a single item auction with multiple bidders, the Vickrey auction allocates the item to the highest bidder and charges them the second highest bid.

When all bidders' values are i.i.d distributed, the Vickrey auction with Myerson Reserve ($r = \phi^{-1}(0)$) gets optimal revenue in expectation Hartline and Roughgarden (2009).

## E.2 DIFFERENTIAL PRIVACY BASICS

We present the definition of pure DP and approximate DP below.

**Definition E.3** (Differential privacy). An algorithm $\mathcal{A} : \mathbb{R}_+^n \to \mathbb{R}$ is $(\epsilon, \delta)$-*approximate* DP if for neighboring dataset $V, V' \in \mathbb{R}_+^n$ that differs in only one data point, and any possible output $O$, we have: $\Pr[\mathcal{A}(V) = O] \leq \exp(\epsilon) \Pr[\mathcal{A}(V') = O] + \delta$. We say it satisfies *pure* DP for $\delta = 0$.

A key property we leverage from differential privacy is its immunity to post-processing. Post-processing refers to any computation or transformation applied to the output of a DP algorithm after the data has been privatized. In our context, Myerson's auction can be seen as a post-processing step. Therefore, applying Myerson's auction to a differentially private release of the empirical distribution preserves the original privacy guarantees of the input distribution.

**Lemma E.4** (Immunity to Post-Processing). *Let $\mathcal{A} : \mathbb{R}_+^n \to \mathbb{R}$ be an $(\epsilon, \delta)$-DP algorithm, and let $f : \mathbb{R} \to \mathbb{R}$ be a random function. Then, $f \circ \mathcal{A} : \mathbb{R}_+^n \to \mathbb{R}$ is also $(\epsilon, \delta)$-DP.*

**Lemma E.5** (Basic Composition Theorem (Dwork et al., 2006)). *Let $\mathcal{M}_1 : \mathcal{D} \to \mathcal{R}_1$ and $\mathcal{M}_2 : \mathcal{D} \to \mathcal{R}_2$ be two mechanisms that are $(\varepsilon_1, \delta_1)$-differentially private and $(\varepsilon_2, \delta_2)$-differentially private, respectively. Then, the composition of $\mathcal{M}_1$ and $\mathcal{M}_2$, denoted as $(\mathcal{M}_1, \mathcal{M}_2) : \mathcal{D} \to (\mathcal{R}_1 \times \mathcal{R}_2)$, satisfies $(\varepsilon_1 + \varepsilon_2, \delta_1 + \delta_2)$-differential privacy.*

## E.3 PROBABILITY INEQUALITIES

Next, we present the Dvoretzky–Kiefer–Wolfowitz(DKW) inequality that, upper bound the probability that the empirical CDF differs from the CDF of the true distribution.

**Lemma E.6** (DKW Inequality (Dvoretzky et al. (1956))). *Let $X_1, X_2, \ldots, X_m$ be i.i.d random variables with cumulative distribution function $F(\cdot)$. Let $F_m$ denote the associated empirical distribution function defined as $F_m(x) = \frac{1}{m} \sum_{i \in [m]} \mathbb{1}_{\{X_i \leq x\}}$. Then, we have the following probability bound:*

$$\Pr[\max_{x \in \mathbb{R}} |F_m(x) - F(x)| > \epsilon] \leq 2 \exp(-2m\epsilon^2)$$

## E.4 STOCHASTIC DOMINANCE BASICS

Next, we introduce some technical preliminaries on the distance and dominance between distributions.

**Definition E.7** (First Order Stochastic Dominance). For distribution $\mathcal{D}$ and $\mathcal{D}'$, we denote the cdf of them as $F_{\mathcal{D}}, F_{\mathcal{D}'}$, respectively. Distribution $\mathcal{D}$ first order stochastically dominates distribution $\mathcal{D}'$ if:

- For any outcome $x$, $F_{\mathcal{D}(x)} \leq F_{\mathcal{D}'}(x)$.

- For some $x$, $F_{\mathcal{D}(x)} < F_{\mathcal{D}'}(x)$

---

[10]If there are multiple bidders with highest virtual value, break ties arbitrarily, e.g., in lexicographical order

[11]the max of the value at which her virtual value is zero and the value at which her virtual value becomes largest

We denote $\mathcal{D} \succeq \mathcal{D}'$ for $\mathcal{D}$ first order stochastically dominates $\mathcal{D}'$. For product distribution $\mathbf{D}$ and $\mathbf{D}'$, if for every $i$, $\mathcal{D}_i \succeq \mathcal{D}'_i$, we say that $\mathbf{D} \succeq \mathbf{D}'$.

**Definition E.8** (Kolmogorov-Smirnov distance). For probability distributions $\mathcal{D}_1, \mathcal{D}_2$ on $\mathbb{R}$, and let $F_1$, $F_2$ denote the culmulative function of $\mathcal{D}_1, \mathcal{D}_2$. Then, the Kolmogorov-Smirnov distance of $\mathcal{D}_1$ and $\mathcal{D}_2$ is defined as follows:

$$d_{ks}(\mathcal{D}_1, \mathcal{D}_2) = \sup_{x \in \mathbb{R}} |F_1(x) - F_2(x)|.$$

Moreover, we call $\mathcal{D}_1$ and $\mathcal{D}_2$ $t$-close if $d_{ks}(\mathcal{D}_1, \mathcal{D}_2) \leq t$.

# F RESULTS FROM BAYESIAN MECHANISM DESIGN

## F.1 REVENUE LOSS

We first state a lemma that guarantee the expected revenue loss by additive discretization by $\epsilon$ is upper bounded by $\epsilon$:

**Lemma F.1** (Additive Discretization of Value Space (Lemma 6.3 in arXiv version of Devanur et al. (2016))). *Given any product distribution $\mathbf{D}$ and $\mathbf{D}'$, where $\mathbf{D}'$ is obtained by rounding down the values from $\mathbf{D}$ to the closest multiples of $\epsilon$, we have:*

$$\mathsf{OPT}(\mathbf{D}') \geq \mathsf{OPT}(\mathbf{D}) - \epsilon$$

**Lemma F.2** (Weak Revenue Monotonicity (Devanur et al., 2016)). *Suppose $\mathsf{D}, \mathsf{D}'$ be two product distribution such that $\mathsf{D}'$ is first order stochastic dominated by $\mathsf{D}$, then the optimal revenue for these distributions satisfies the following:*

$$\mathsf{OPT}(\mathsf{D}) \geq \mathsf{OPT}(\mathsf{D}')$$

**Lemma F.3** (Strong Revenue Monotonicity (Devanur et al., 2016)). *Let $\mathsf{D}'$ be a product distribution with finite support. There exists a mechanism $M_0$ such that $M_0$ is an optimal auction for $\mathsf{D}$, and for all finite support distributions $\mathsf{D} \succeq \mathsf{D}'$:*

$$\mathsf{Rev}(M_0, \mathsf{D}) \leq \mathsf{Rev}(M_0, \mathsf{D}')$$

## G  REVENUE SHIFT THEOREM

In this subsection, we introduce technicals to show our revenue shift theorem. We present the definition of an increasing function w.r.t vector input below.

**Definition G.1** (Increasing Functions ). Let $u : \mathbb{R}^n \to \mathbb{R}$, we say that $u$ is increasing if for every $\mathbf{v} = (v_1, \ldots, v_k), \mathbf{v}' = (v'_1, \ldots, v'_k)$ such that $v'_i \geq v_i$, it holds that $u(\mathbf{v}') \geq u(\mathbf{v})$.

We now extend this definition to the scenario where the input vectors follow *distributions*. In this case, the difference in the expected output on these vectors can be bounded by a function of the statistical distance between their underlying distributions.

**Lemma G.2** (Utility Difference for Bounded Distribution (Guo et al., 2021)). *Let* $\mathsf{D} = \mathcal{D}_1 \times \ldots \times \mathcal{D}_k$, $\mathsf{D}' = \mathcal{D}'_1 \times \ldots \times \mathcal{D}'_k$ *be product $k$-dimensional distributions with $d_{ks}(\mathcal{D}_i, D'_i) \leq \alpha_i$. Then for every increasing function $u : \mathbb{R}^k \to [0, \bar{u}]$, it holds that:*

$$|\mathbb{E}_{\mathbf{v} \sim \mathsf{D}}[u(\mathbf{v})] - \mathbb{E}_{\mathbf{v}' \sim \mathsf{D}'}[u(\mathbf{v}')]| \leq \bar{u} \cdot (\sum_{j=1}^{n} \alpha_i)$$

Our proof of the revenue shift theorem relies on the property that the optimal revenue (as characterized by Myerson in our setting) equals the maximum payment achievable from a given value profile and is an increasing function of the observed bids. The formal proof is provided below:

**Theorem G.3** (Revenue Shift). *Given two product distribution $\mathbf{D} \succeq \mathbf{D}'$ whose valuations are bounded by $[0, h]$, with $d_{ks}(\mathbf{D}_i, \mathbf{D}'_i) \leq \alpha_i$ for any bidder/entry $i$, the optimal revenue of these distribution satisfies:*

$$0 \leq \mathbb{E}[\mathsf{Rev}(M_\mathbf{D}, \mathbf{D}) - \mathsf{Rev}(M_{\mathbf{D}'}, \mathbf{D}')] \leq (\sum_{i \in [k]} \alpha_i)h$$

*Proof.* According to weak revenue monotonicity F.2, we get that the optimal revenue of a distribution $\mathcal{D}$ over support $[0, h]$ is an increasing function defined in Def. G.1, and the optimal revenue is upper bounded by $h$. Hence, by lemma G.2, we get that for any distribution $\mathbf{D} \succeq \mathbf{D}'$.

$$0 \leq \mathbb{E}[\mathsf{Rev}(M_\mathbf{D}, \mathbf{D}) - \mathsf{Rev}(M_{\mathbf{D}'}, \mathbf{D}')] \leq (\sum_{i \in [k]} \alpha_i)h$$

$\square$

## H   MORE DETAILS FOR BOUNDED DISTRIBUTIONS

### H.1   PRIVATE QUANTILE ESTIMATION

It's worthwhile to present and state the quantile estimation oracle below. In our paper, we apply a similar quantile estimation algorithm as in Kaplan et al. (2022), which applies the exponential mechanism (Smith, 2011) efficiently on the dataset/samples.

---

**Algorithm 4** DP Quantile, Bounded Distribution DPQUANT (Kaplan et al., 2022)

---

**Input:** $n$ samples $V = \{v_1, \ldots, v_n\}$, range $[lb, ub]$, set of quantiles $Q := \{q_1, \ldots, q_m\}$, privacy parameter $\epsilon_p$, DP oracle $\mathcal{A}$ that estimate a single quantile with privacy $\epsilon_p/(\log_2 m + 1)$ (Alg. 5).
 1: Rank the quantiles $Q$ in increasing order.
 2: **if** $m == 1$ **then return** $\{\mathcal{A}((lb, ub), V, q_1)\}$
 3: **end if**
 4: $s \leftarrow \mathcal{A}((lb, ub), V, q_{\lfloor m/2 \rfloor})$.
 5: Separate the samples $V$ by $s$ into left and right quantiles, i.e., $V_l := \{v < s | v \in V\}$, $V_r := \{v > s | v \in V\}$.
 6: Update the candidate quantile into $Q_l$ and $Q_r$, where $Q_l := \{q < q_{\lfloor m/2 \rfloor} | q \in Q\}$, $Q_r = \{q > q_{\lfloor m/2 \rfloor} | q \in Q\}$
 7: **return** $\{\text{DPQUANT}(V_l, (lb, s), Q_l, \epsilon_p, \mathcal{A})\} \cup \{s\} \cup \{\text{DPQUANT}(V_r, (s, ub), Q_r, \epsilon_p, \mathcal{A})\}$

---

We define the utility function measuring the accuracy of a given quantile estimation $I$ of quantile $q$. This function measures the number of points between the true value of a given quantile $q$ and its estimation $I$.

**Definition H.1** (Utility Function of Quantile Estimation (Smith, 2011)). Given a dataset $V \in [lb, ub]^n$, a quantile $q$, and an estimation $I$, the utility function of quantile estimation is defined as:

$$u(V, I, q) := -|\{v \in V | v < I\} - \lfloor q \cdot n \rfloor|$$

For multiple quantiles $Q = (q_1, \ldots, q_m)$ and estimation $I = (I_1, \ldots, I_m)$, the utility is defined as the worst utility of these quantile estimations, i.e.,

$$u(V, I, Q) := \min_{r \in [m]} u(V, I_r, q_r)$$

Here we present the DP Single Quantile estimation below. WE apply this algorithm in thhe DP Quantile with privacy $\epsilon_p/(\log_2 m + 1)$, for number of quantiles $m$ and required (pure) privacy $\epsilon_p$. The DP Single Quantile guarantees pure privacy by efficiently implementing the exponential algorithm with the utility we previously described. For DP Single Quantile, we use the convention that for $v_k = v_{k-1}$, the sample probability is 0, under this convention, only interval with *positive* length will have the probability to be sampled.

---

**Algorithm 5** DP Single Quantile

---

**Input:** $n$ samples $V = \{\mathbf{v}_1, \ldots, \mathbf{v}_n\}$, range $[lb, ub]$, quantile $q$, privacy parameter $\epsilon_p$
 1: Rank samples in $V$ in increasing order.
 2: **for** $i = 1 \rightarrow n + 1$ **do**
 3:     Let interval $I_i = [v_{i-1}, v_i]$, where $x_0 = lb$, $x_{n+1} = ub$.
 4: **end for**
 5: Sample an interval $I_k$ from this set of intervals, with probability $\exp(\epsilon_p u(V, I_k, q)/2) \cdot (v_k - v_{k-1})$.
 6: **return** a uniformly random point from interval $I_k$.

---

Now we present the utility for single quantile for duplicates value below. This proof is similar to the one adopted in the Appendix A.1 of Kaplan et al. (2022).

**Lemma H.2** (Utility for Single Quantile, Discrete Distribution). *With probability $1 - \delta$, given samples $V \in [0, h]^n$ and quantile $q \in [0, 1]$, where samples may have the same value, but those with different*

*values will differ by at least $\epsilon_a$. Then, the exponential mechanism described in Alg. 5 would output $s$ with $(\epsilon_p, 0)$-DP and:*

$$|u(V, I, q)| \leq 2 \cdot \frac{\log \phi - \log \delta}{\epsilon_p}.$$

*where $\phi := h/\epsilon_a$.*

*Proof.* Let $I_t$ be an interval that $u(V, I_t, q) \leq -\gamma$. Then the probability that we sample a point from $I_t$ is at most:

$$\Pr[\mathcal{A}(V) = I_t] \leq \frac{\exp(-\epsilon_p \gamma / 2)(v_i - v_{i-1})}{\sum_{i \in [n], x_i \neq x_{i-1}} \exp(\epsilon_p u(V, I_i, q)) \epsilon_a}$$

$$\leq \frac{\exp(-\epsilon_p \gamma / 2) h}{\exp(\epsilon_p u(V, I_o, q)) \epsilon_a}$$

$$\leq \phi \exp(-\epsilon_p \gamma / 2)$$

where $\mathcal{A}$ denotes the output by DP Single Quantile, and $o$ denote the optimal interval with for quantile $q$ that has zero utility. Next, it follows that with probability less than $\delta$, the returned interval will have a utility at most $-\gamma$ for $\gamma = \Theta(\log \phi + \log(1/\delta))/\epsilon_p$, which completes the proof. $\square$

We have proven that the single quantile algorithm still holds similar accuracy guarantee under our assumption on the dataset which the data points can have same value, but if they are different, then their values will differ by at least $\epsilon_a$. Following the same logic as in Kaplan et al. (2022), we can also demonstrate the accuracy guarantee of the quantile estimation algorithm in our distribution setting.

**Lemma H.3** (Utility of DP Quantile (Thm. 3.3 in Kaplan et al. (2022))). *With probability $1 - \delta$, given samples $V \in [0, h]^n$ and quantile $Q = (q_1, \ldots, q_m)$ and privacy parameter $\epsilon_p$, where samples may have the same value, but those with different values will differ by at least $\epsilon_a$. Then, the Alg 4 will output $S = (s_1, \ldots, s_m)$ with $(\epsilon_p, 0)$-DP, such that:*

$$\text{ERR}(V, S, Q) := -u(V, S, Q) \leq 2(\log m + 1) \cdot \frac{\log \phi + \log m - \log \delta}{\epsilon_p}$$

*where $\phi := h/\epsilon_a$.*

## H.2 REVENUE GUARANTEE OF PRIVATE MYERSON

We now present the complete proof of the accuracy guarantee for the private Myerson mechanism under the bounded distribution setting.

**Theorem H.4** (Revenue Guarantee of Private Myerson (Alg. 1), formal version of Theorem 3.2). *Given $n$ samples $\widehat{V} \in [0, h]^{k \times n}$ of the joint distribution $\mathbf{D}$, DPMYER (Alg. 1) is $(2k\epsilon_p, 0)$-DP, and the expected revenue of this mechanism is close to the optimal revenue of distribution $\mathbf{D}$, i.e., with probability $1 - \delta$:*

$$|\mathbb{E}[\text{Rev}(M_{\text{DPMYER}}, \mathbf{D}) - \text{Rev}(M_\mathbf{D}, \mathbf{D})]| \leq \widetilde{\Theta}((\epsilon_q + \frac{1}{\epsilon_p \cdot n} + \frac{1}{\sqrt{n}}) \cdot kh + \epsilon_a).$$

*for $n \geq \max\{2 \log(4k/\delta)/\epsilon_q^2, 8(\log(1/\epsilon_q) + 1)(\log(hk \log(1/\epsilon_q)/(\epsilon_a \delta)))/(\epsilon_q \epsilon_q)\}$, which can be further simplifies to $n = \widetilde{\Omega}(\max\{1/(\epsilon_p \epsilon_q), 1/\epsilon_q^2\})$. Furthermore, under $\epsilon_a = \epsilon_q = \epsilon$, and that $n = \Theta(\max\{\epsilon^{-2} \log(k/\delta), \epsilon^{-2} \log(1/\epsilon) \log(\frac{hk \log(2/\epsilon)}{\epsilon \delta}), \epsilon^{-2} \log(k/\delta)\})$, which can be further simplifies to $n = \widetilde{\Theta}(\epsilon^{-2})$, we have:*

$$|\mathbb{E}[\text{Rev}(M_{\text{DPMYER}}, \mathbf{D}) - \text{OPT}(\mathbf{D})]| \leq \widetilde{\Theta}((\epsilon + \epsilon^2/\epsilon_p)kh).$$

*Proof.* **Privacy** We know that the quantile estimates from DPQE is $(\epsilon_p, 0)$ private (Lem. H.2). Since DP is immune to post-processing (Lem. E.4), and that the output of allocation and payment combination is $2k$ dimensional, by composition theorem (Lem. E.5), our algorithm is $(2k\epsilon_p, 0)$-DP.

We include all distributions considered in this proof in Figure 1 below.

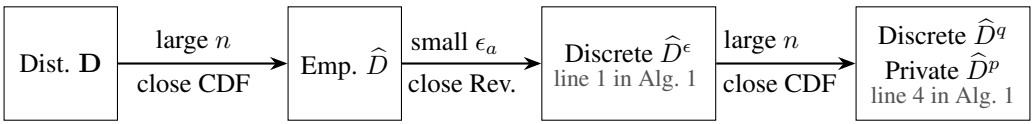

Figure 4: **Distribution analyzed for DPMYER(Alg. 1)**. We establish connections between the accuracy/revenue guarantee of the original distribution $\mathbf{D}$ with the empirical distribution $\widehat{D}$, the value-discretized $\widehat{D}^\epsilon$, the quantile-discretized $\widehat{D}^q$ and the distribution $\widehat{D}^p$ returned by DPQUANT(Alg. 4).

From the DKW inequality E.6, we know that with probability $1 - \delta_1$, the cumulative density function of the empirical distribution is close to the true distribution, i.e., for all attribute $i \in [k]$,

$$d_{\mathrm{ks}}(\mathbf{D}_i, \widehat{D}_i) := \max_v |(F_{\mathbf{D}}(v) - F_{\widehat{D}}(v))| \leq \sqrt{log(2k/\delta_1)/2n}.$$

We condition on this event holds since after discretization by $\epsilon_a$, each value at most decreases by $\epsilon_a$ in the new distribution $\widehat{D}^\epsilon$. By Lemma F.1, we note that the optimal revenue at most decreases by $\epsilon_a$:

$$\mathbb{E}[\mathsf{Rev}(M_{\widehat{D}}, \widehat{D}) - \mathsf{Rev}(M_{\widehat{D}^\epsilon}, \widehat{D}^\epsilon)] \leq \epsilon_a.$$

Next, we discretize again on distribution $\widehat{D}^\epsilon$, which additively discretizes this distribution in the quantile space. We denote this distribution as $\widehat{D}^q$. Consequently, for any attribute $i \in [k]$, we have:

$$d_{\mathrm{ks}}(\widehat{D}^\epsilon, \widehat{D}^q) \leq \epsilon_q.$$

Next, from Lemma H.3, we get that with probability $1 - \delta_2$, for each attribute $i \in [k]$, the following holds:

$$\begin{aligned} d_{\mathrm{ks}}(\widehat{D}_i^q, \widehat{D}_i^p) &= |-\mathsf{ERR}(V, S, Q)|/n \\ &\leq 2(\log m + 1) \cdot \frac{\log h - \log \epsilon_a + \log m + \log k - \log \delta_2}{\epsilon_p \cdot n} := \widehat{\epsilon}. \end{aligned}$$

Next, we show by $n \geq \max\{2\log(4k/\delta)/\epsilon_q^2, 4(\log m + 1)(\log(hmk/(2\epsilon_a\delta)))/(\epsilon_q\epsilon_q)\}$[12], we have $\mathbf{D} \succeq \widehat{D}^q$: 1) For $n \geq 2\log(4k/\delta)/\epsilon_q^2$, we have that $d_{\mathrm{ks}}(\mathcal{D}_i, \widehat{D}_i) \leq \epsilon_q/2$. 2) For $n \geq 4(\log m + 1)(\log(hmk/(2\epsilon_a\delta)))/(\epsilon_q\epsilon_q)$, we have that $\widehat{\epsilon} \leq \epsilon_q/2$. Since quantile discretization shift the distribution down by $[\epsilon_q, 2\epsilon_q]$, and that additive discretization only shift the distribution to a one that is dominated by it, we have that $\widehat{D}^q$ is still dominated by $\mathbf{D}$.

We condition on both events holding and denote the KS-distance upper bound between the privacy estimation vs the ground truth of the discretized distribution of one attribute as $\widehat{\epsilon}$. Thus, we get that with probability $1 - \delta_1 - \delta_2$, we have the following revenue bound of the final mechanism:

$$\begin{aligned} 0 &\geq \mathbb{E}[\mathsf{Rev}(M_{\widehat{D}^p}, \mathbf{D}) - \mathsf{Rev}(M_{\mathbf{D}}, \mathbf{D})] \\ &\geq \mathbb{E}[\mathsf{Rev}(M_{\widehat{D}^p}, \mathbf{D}) - \mathsf{Rev}(M_{\widehat{D}^p}, \widehat{D}^p)] - |\mathsf{Rev}(M_{\widehat{D}^p}, \widehat{D}^p) - \mathsf{Rev}(M_{\mathbf{D}}, \mathbf{D})| \\ &\geq -|\mathsf{Rev}(M_{\widehat{D}^p}, \widehat{D}^p) - \mathsf{Rev}(M_{\widehat{D}^q}, \widehat{D}^q)| - |\mathsf{Rev}(M_{\widehat{D}^q}, \widehat{D}^q) - \mathsf{Rev}(M_{\mathbf{D}}, \mathbf{D})| \\ &\geq -k\widehat{\epsilon}h - |\mathsf{Rev}(M_{\widehat{D}^q}, \widehat{D}^q) - \mathsf{Rev}(M_{\widehat{D}^\epsilon}, \widehat{D}^\epsilon)| - |\mathsf{Rev}(M_{\widehat{D}^\epsilon}, \widehat{D}^\epsilon) - \mathsf{Rev}(M_{\mathbf{D}}, \mathbf{D})| \\ &\geq -k(\widehat{\epsilon} + \epsilon_q)h - |\mathsf{Rev}(M_{\widehat{D}^\epsilon}, \widehat{D}^\epsilon) - \mathsf{Rev}(M_{\widehat{D}}, \widehat{D})| - |\mathsf{Rev}(M_{\widehat{D}}, \widehat{D}) - \mathsf{Rev}(M_{\mathbf{D}}, \mathbf{D})| \\ &\geq -k(\widehat{\epsilon} + \epsilon_q)h - \epsilon_a - \sqrt{log(2k/\delta_1)/2n}kh, \end{aligned}$$

where the first inequality follows from the optimality of mechanism $M_{\mathbf{D}}$ on distribution $\mathbf{D}$, and that $\mathbf{D} \succeq \widehat{D}^q$ by our choice of $n$. The second inequality follows from rearranging the term. The

---

[12]The quantity in the theorem statement is $8(\log(1/\epsilon_q) + 1)(\log(hk \log(1/\epsilon_q)/(\epsilon_a\delta)))/(\epsilon_q\epsilon_q)$ and is greater than the second term in the max here.

third inequality follows from strong revenue monotonicity F.3 and that $\mathbf{D} \succeq \widehat{D}^p$, we get that the term $\mathbb{E}[\mathsf{Rev}(M_{\widehat{D}^p}, \mathbf{D}) - \mathsf{Rev}(M_{\widehat{D}^p}, \widehat{D}^p)] \geq 0$. The next few inequalties follows from applying the revenue shift theorem (Thm G.3) iteratively for: 1) $\widehat{D}^p$ and $\widehat{D}^q$ with distance $\widehat{\epsilon}$, 2) $\widehat{D}^\epsilon$ and $\widehat{D}^q$ with distance $\epsilon_q$, and 3) $\widehat{D}$ and $\mathbf{D}$ with distance $\sqrt{\log(2k/\delta_1)/2n}$. We also apply Lemma F.1 to upper bound the revenue loss from additive discretization.

Next, we plug in the value of $\widehat{\epsilon}$ and $m = \lfloor 1 + 1/\epsilon_q \rfloor$ to upper bound the value of $\widehat{\epsilon}$, i.e.,

$$\widehat{\epsilon} \leq \frac{4}{\epsilon_p \cdot n} \cdot \log(\frac{1}{\epsilon_q}) \cdot (\frac{mhk}{\epsilon_a \cdot \delta_2}).$$

Finally, letting $\delta_1 = \delta_2 = \delta/2$, we get the final result: with probability $1 - \delta$:

$$| \mathbb{E}[\mathsf{Rev}(M_{\widehat{D}^p}, \mathbf{D}) - \mathsf{Rev}(M_\mathbf{D}, \mathbf{D})]|$$
$$\leq (\epsilon_q + \frac{10}{\epsilon_p \cdot n} \log(\frac{1}{\epsilon_q}) \cdot \log(\frac{hk}{\epsilon_q \epsilon_a \cdot \delta}) + \sqrt{\frac{\log(4k/\delta)}{2n}}) \cdot kh + \epsilon_a$$
$$\leq \widetilde{\Theta}((\epsilon_q + \frac{1}{\epsilon_p \cdot n} + \frac{1}{\sqrt{n}}) \cdot kh + \epsilon_a),$$

where the $\widetilde{\Theta}$ hide the polylog factors. Furthermore, let $\epsilon_q = \epsilon_a$ and let $n \geq \epsilon^{-2} \log(2k/\delta_1)/2$ give us the following bound:

$$| \mathbb{E}[\mathsf{Rev}(M_{\widehat{D}^p}, \mathbf{D}) - \mathsf{Rev}(M_\mathbf{D}, \mathbf{D})]| \leq \widetilde{\Theta}((\epsilon + \epsilon^2/\epsilon_p)kh).$$

Next, we let $\delta$ in the statement to be $1/k$ of the $\delta$ we used in this proof. This $\delta$ would only affect the revenue by $\mathrm{poly} \log$ factors hence are hidden in the $\widetilde{\Theta}$. □

### H.3 RUNNING TIME FOR PRIVATE MYERSON

**Theorem H.5** (Running time, DP Myerson for Bounded Distribution). *Given the same parameters as stated in Theorem 3.2, the running time of* DPMYER *(Alg.1) is* $\Theta(\log(1/\epsilon_q)n + kn) = \widetilde{\Theta}(kn)$ *and requires* $\Theta(\log(1/\epsilon_q) = \widetilde{\Theta}(1)$ *passes of the samples.*

*Proof.* We describe below how to implement DPMYER efficiently.

- For the additive discretization step, we rounded down each value to the closest multiples of $\epsilon_a$. This step runs in $O(kn)$ time and requires 1 pass of the dataset.

- For quantile preparation, this step takes $\lfloor 1/\epsilon_q \rfloor + 1$ time.

- For the DP quantile estimation step, we know that it requires $\log(\lfloor 1/\epsilon_q \rfloor + 1)$ passes of the dataset. The running time of each of these pass depends not on $n$, but on the number of distinct value we have after additive discretization (i.e., $O(h/\epsilon_a)$), and the number of quantiles we want to calculate our utility (Def. H.3) on (i.e., $O(1/\epsilon_q)$). Each pass of the distribution will take $O(k/(\epsilon_a \epsilon_q))$ time. Since $\epsilon_a = \epsilon_q = \epsilon$, we have that the total running time is $\widetilde{\Theta}(k\epsilon^{-2}) = \widetilde{\Theta}(kn)$.

Summing them up provides the final guarantee. □

# I    MORE DETAILS FOR UNBOUNDED DISTRIBUTIONS

## I.1    TRUNCATION POINT LEMMA

Here we present a lemma on how to truncate the regular distribution. Notice that this truncation point depends on the optimal revenue produced by the distribution itself. In order to estimate this truncation point up by approximation, an approximation on the revenue is needed.

**Lemma I.1** (Truncation of Regular Distribution (Devanur et al., 2016))**.** *For any product regular distribution* $\mathbf{D} = (D_1, \ldots, D_k)$*, given any* $\epsilon \in (0, 1/4]$*, let* $\bar{v} \geq \frac{1}{\epsilon}\text{OPT}(\mathbf{D})$ *be the truncation point, and let* $\bar{D}_1, \ldots, \bar{D}_k$ *be the distribution after truncating* $\mathbf{D}$ *by point* $\bar{v}$*. Then, we have*

$$\text{OPT}(\bar{\mathbf{D}}) \geq (1 - 2\epsilon)\text{OPT}(\mathbf{D}).$$

## I.2    THE EMPIRICAL RESERVE ALGORITHM

In this section, we formally introduce the details of *Empirical Reserve* algorithm, and how it approximates the optimal revenue when there is only one bidder. We run a $\delta/2$-guarded reserve mechanism to collect an estimation on the optimal revenue of product distribution $\mathbf{D}$. Then, we analyze the approximation guarantee, the incentive robustness and the convergence of this algorithm. In this subsection, the quantile $q$ is defined as the value corresponds to the top $q$ quantile as opposed to in out context, the quantile is defined as the bottom $q$ quantile. First, we describe the $\beta$-guarded empirical reserve algorithm (Alg. 6).

---

**Algorithm 6** Empirical Reserve ER (Huang et al., 2018b)

---

**Parameters:** distribution $\mathcal{D}$, failure probability $\delta$, guarded parameter $\beta$, accuracy parameter $\epsilon_{\text{ER}}$.
**Input:** $m = \Theta(\beta^{-1}\epsilon^{-2}\log(\beta^{-1}\epsilon^{-1})\log(1/\delta_{\text{ER}}))$ samples from distribution $\mathcal{D}$
  1: Sort $m$ samples in the decreasing order, i.e., $v_1 \geq v_2 \geq \ldots \geq v_m$.
  2: Find the smallest index $j \in [\beta \cdot m, m]$ that maximizes the empirical revenue, i.e.,

$$j = \arg \max_{\beta m \leq i \leq m} i \cdot v_i$$

  3: $r_{\mathcal{D}} \leftarrow v_j$                                    ▷ $\beta$-guarded empirical reserve
  4: $R_{\mathcal{D}} \leftarrow j \times v_j/m$                               ▷ Empirical revenue
**Output:** $r_{\mathcal{D}}, R_{\mathcal{D}}$

---

**Definition I.2** ($\beta$-guarded reserve)**.** Given $m$ samples $v_1 \geq v_2 \geq \cdot \geq v_m$, the *empirical reserve* is

$$\arg \max_{i \geq 1} i \cdot v_i$$

If we only consider $i \geq \beta m$ for some parameter $\beta$, it is called the $\beta$-guarded empirical reserve.

**Lemma I.3** (Empirical Reserve, $\gamma$-strongly-Regular, Thm 3.3 in Huang et al. (2018b))**.** *The empirical reserve with* $m = \Theta(\epsilon^{-3/2}\log(\epsilon^{-1}))$ *samples is* $(1 - \epsilon)$*-approximate for all* $\gamma$*-strongly regular distributions, for a constant* $\alpha > 0$*.*

**Lemma I.4** (Optimal Quantile)**.** *Let* $q$ *denote the quantile,* $v(q)$ *denote the value of that quantile, and let* $R(q) = qv(q)$ *be thr revenue as a function over the quantile space. Let* $q^*$ *and* $v^* = v(q^*)$ *be the revenue-optimal quantile and reserve price respectively. Then,*

- *(Hartline et al. (2008)) For every MHR distribution,* $q^* \geq \frac{1}{e}$*.*

- *(Cole and Roughgarden (2014)) For any* $\gamma$*-strongly regular distribution,* $q^* \geq \gamma^{\frac{1}{1-\gamma}}$*.*

**Lemma I.5** (Empirical Reserve, Bounded, Thm 3.6 in Huang et al. (2018b))**.** *The empirical reserve with* $m = \Theta(H\epsilon^{-2}\log(H\epsilon^{-1}))$ *samples is* $(1 - \epsilon)$*-approximate for all distributions with support* $[1, H]$*.*

**Lemma I.6** ($\beta_0/2$-guarded empirical reserve, Thm. 3.5 in Arxiv Version of Huang et al. (2018b))**.** *The* $\frac{\beta_0}{2}$*-guarded empirical reserve with* $m = \Theta(\beta_0^{-1}\epsilon^{-2}\log(\beta_0^{-1}\epsilon^{-1}))$ *gives revenue at least* $(1 - \epsilon)R_{\beta_0}^*$ *for all distributions, where* $R_{\beta_0}^*$ *is the optimal revenue by prices with sale probability at least* $\beta_0$*, in expectation.*

## I.3 DIFFERENTIAL PRIVATE QUANTILE ESTIMATION FOR UNBOUNDED DISTRIBUTION

For the sake of completeness, we present the algorithm for DP quantile estimation for unbounded distribution (Durfee, 2023) below. This algorithm works for value distributions with lower bound and no upper bound. Intuitively, the algorithm gradually guess the correct value for the quantile $q$, and check whether there are sufficient number of points in the given dataset $V$ that are below the guessed value. Specifically, for $\beta > 1$, the algorithm first check whether $\beta - 1$ is the correct quantile, then scale this guess roughly exponentially to $\beta^2 - 1, \beta^3 - 1, \ldots, \beta^i - 1$ until the algorithm stops. This algorithm guarantees that with high probability, the value it finds has the quantile greater than $q - \Delta$ for some small $\Delta$. This Alg. 7 can be run in time in $\widetilde{O}(n)$ by specific dictionary data structure(Section 4.2 of Durfee (2023)), where the $\widetilde{O}$ hides the log factors of $n$ and target value $\widehat{v}$.

---

**Algorithm 7** DP Quantile Estimation for Unbounded Distribution, DPQUANTU$(V, Q)$ (Durfee, 2023)

---

**Input:** $n$ samples $V = \{v_1, \ldots, v_n\}$, quantile $q$, privacy parameter $\epsilon_p$, parameter $\beta > 1$
1: Let $\widehat{T} = qn + \text{EXP}(2/\epsilon_p)$.
2: **for** $i = 1, 2, 3, \ldots$ **do**
3:     $f_i(V) \leftarrow |\{v_j \in V | v_j + 1 < \beta^i\}|$
4:     $\epsilon_i \leftarrow \text{EXP}(2/\epsilon_p)$.
5:     **if** $f_i(V) + \epsilon_i \geq \widehat{T}$ **then**   Output $\beta^i - 1$ and halt
                                              $\triangleright$ Check whether current $\beta^i - 1$ exceeds the threshold.
6:     **end if**
7: **end for**

---

**Lemma I.7** (DP Guarantee of DP Quantile, Unbounded (Durfee, 2023)). *Alg. 7 is $\epsilon_p$-DP.*

**Definition I.8** (Exponential Noise). The exponential distribution with parameter $z$, i.e., $\text{EXP}(z)$ has the following PDF: For $v \geq 0, f_{\text{EXP}(z)}(v) = \frac{1}{z} \exp(-v/z)$.

We derive the accuracy guarantee of DP Quantile for unbounded distribution below.

**Lemma I.9** (Accuracy Guarantee of DP Quantile, Unbounded). *Given $n$ samples $V = \{v_1, \ldots, v_n\}$, quantile $q$, privacy parameter $\epsilon_p$ and parameter $\beta$. Alg. 7 will output $\widehat{S}$ satisfies the following:*

- ***Unbiased**: $\mathbb{E}[f_{i-1}(V)] \leq qn \leq \mathbb{E}[f_i(V)]$, for $i$ is the iteration of halt.*

- ***Asymptotic $\beta$ approximation**: With probability $1 - 2\delta$:*
$$F((\widehat{S} + 1)/\beta - 1) - \epsilon_p \cdot \log(1/\delta)/2n \leq F(S) \leq F(\widehat{S}) + \epsilon_p \cdot \log(1/\delta)/2n$$
*where $S$ is the value of quantile $q$ and $F(\cdot)$ is the CDF of samples $V$.*

*Proof.* We prove the unbiasedness and approximation guarantee below:

- **Unbiased**: We plug in the value of $\widehat{T}$ into the If statement, i.e., $f_i(V) + \epsilon_i \geq qn + \epsilon$, where $\epsilon$ and $\epsilon_i$ follows the exponential distribution with parameter $2/\epsilon_p$. Then after taking expectations, we have $f_i(V) \geq qn$. The same logic applies to $f_{i-1}(V)$.

- **Asymptotic $\beta$-approximation**: This follows from the tail bound of exponential distribution:
$$\Pr[\epsilon \geq x] = \exp(-2x/\epsilon_p)$$
Reorganizing we have, with probability $1 - \delta$:
$$\epsilon \leq \epsilon_p \log(1/\delta)/2$$
Since $\epsilon_i, \epsilon \geq 0$, we have:
$$\Pr[f_i(V) \leq qn - x] \leq \Pr[\epsilon_i \geq x]$$
$$\Pr[f_{i-1}(V) \geq qn + x] \leq \Pr[\epsilon \geq x]$$
The plugging in $f_i(V) = n \cdot F(\widehat{S})$ and $f_{i-1}(V) = n \cdot F(\widehat{S} + 1/\beta - 1)$ gives us the desired result.

$\square$

### I.4 ANALYSIS FOR DPKOPT

In this subsection, we formally present our algorithm for estimate an $\Theta(k)$-approximation of the optimal revenue. We private estimate the maximum revenue (of a single item single bidder setting) from each bidder's distribution. Aggregating these private estimation of the empirical revenue gives as a $\Theta(k)$-approximation of the optimal revenue for the product distribution.

---

**Algorithm 8** DP Estimation for Optimal Revenue DPKOPT($V, \epsilon_q, \epsilon_a, \epsilon_p, \eta$)

**Input:** $n$ samples $V = \{\mathbf{v}_1, \ldots, \mathbf{v}_n\}$, quantile discretization $\epsilon_q$, additive discetization $\epsilon_a$, privacy parameter $\epsilon_p$, regularity parameter $\eta$.
1: **for** $d = 1 \to k$ **do**
2:      $\widehat{q} \leftarrow 1/4 \cdot \eta^{1/(1-\eta)}$
3:      Let $ub_d \leftarrow$ DPQUANTU($V_{[d,:]}, 1 - \widehat{q}, \epsilon_p$).          ▷ Estimate the truncation point of $D_d$.
4:      Truncate distribution $D_d$ at $ub_d$ as $\widehat{D}_d$, and discretize $\widehat{D}_d$ by additive $\epsilon_a$ in the value space.
5:      Prepare the quantile to be estimated, $Q \leftarrow \{1 - \widehat{q}, 1 - \widehat{q} - \epsilon_q, \cdots, 1 - \widehat{q} - \lfloor \frac{1-\widehat{q}}{\epsilon_q} \rfloor \cdot \epsilon_q, 0\}$.
6:      $\widehat{S}_{[d,:]} \leftarrow$ QESTIMATE($Q, V_{[d,:]}, \epsilon_p$)          ▷ Apply DP quantile estimate(Alg. 4).
7:      Let $\widehat{F}_d$ be the distribution generated by value profile $\widehat{S}_{[d,:]}$ and quantile set $Q$.
8:      SREV$_d \leftarrow \max_{r \in \widehat{S}} r(1 - \widehat{F}_d(r))$.          ▷ Estimate the optimal revenue from $\widehat{F}_d$(Alg. 6).
9: **end for**
10: KREV $\leftarrow \sum_{d \in [k]}$ SREV$_d$
11: **return** KREV

---

**Lemma I.10** (Expected Revenue Guarantee of DPKOPT). *Given $\epsilon \in (1, 1/4)$ and $n = \Theta(\epsilon^{-2} \log(\epsilon^{-1}))$ samples $\widehat{V}$ of the joint distribution $\mathbf{D} \in [\mathbf{0}, \mathbf{h}]^\mathbf{k}$, the expected revenue of Myerson fitted under DPKOPT (Alg. 2) over distribution $\mathbf{D}$ is close to the optimal revenue of distribution $\mathbf{D}$, i.e., with probability $1 - k\delta$, for every $i \in [k]$, $|E[\mathsf{Rev}(M_{ER(\mathcal{D}_p)}, \mathcal{D}_p) - \mathsf{OPT}(\mathcal{D}_i)]| \leq \widetilde{O}(k(\frac{\epsilon^2}{\epsilon_p} + \epsilon + \epsilon_q + \epsilon_a))$.*

*Proof.* Since bidders' distributions are independent, we analyze the the revenue guarantee of each SREV$_i$ for $i \in [k]$ separately, and we omitted the subscription on $i$ in the proofs.

We denote $q_0 = \eta^{1/(1-\eta)} = 4\widehat{q}$ as the (top) optimal quantile for the empirical revenue (Lem. I.4). We denote the output value from line 3 as $\widehat{v}_{\max}$.

From Lemma I.9, with probability $1 - \delta_2$, for $n \geq 4\epsilon_p \log(1/\delta_2)/\widehat{q} = \Theta(\epsilon_p \log(1/\delta_2))$ samples:

$$F(\widehat{v}_{\max}) \geq 1 - \widehat{q} - \epsilon_p \log(1/\delta_2)/2n \geq 1 - \frac{3}{8}q_0.$$

This is saying that with high probability, the returned value is greater than that of the top $3q_0/8$ quantile. Thus, conditioned on this event, the revenue from applying the empirical reserve on the truncated distribution (which equals the optimal revenue from the truncated distribution) is equivalent to that of applying the empirical reserve on the original distribution. The revenue generated by empirical reserve equals the expected optimal revenue from the same distribution. Hence, we concluded that this truncation won't affect the optimal revenue.

Next, we privately estimate the pre-specified quantiles of the truncated distribution, and output the revenue generated from it. We denote the output distribution as $\mathcal{D}_p$, and the truncated distribution as $\mathcal{D}_{\mathrm{TR}}$. By similar arguments as in Theorem 3.2, we know that, with probability $1 - \delta_2$, for $n \geq \widetilde{\Theta}(1/\epsilon_p)$:

$$d_{\mathrm{ks}}(\mathcal{D}_p, \mathcal{D}_{\mathrm{TR}}) \leq 2(\log m + 1) \frac{\log \widehat{v}_{\max} - \log \epsilon_a + \log m - \log \delta_2}{\epsilon_p \cdot n} + \epsilon_q$$

$$:= \epsilon_{\mathrm{PTR}} = \widetilde{\Theta}(\frac{1}{\epsilon_p \cdot n})(\leq 1/16q_0)$$

where $m = \lfloor \frac{1-\widehat{q}}{\epsilon_q} \rfloor + 1$. Then, from Thm G.3, the optimal revenue from these distributions differs by at most $\widehat{v}_{\max}\epsilon_{\text{PTR}}$. Notice again from Lemma I.9, with probability $1 - \delta_1$, $\widehat{v}_{\max}\epsilon_{\text{PTR}}$ also satisfies:

$$F(\widehat{v}_{\max}/\beta) \leq 1 - q_0/8$$

This results in $\widehat{v}_{\max} \leq \beta \cdot C_0$, where $C_0$ is the true value of quantile $1 - 1/8q_0$, and can be treated as a constant since $q_0$ is a constant. Aggregating these together gives us the optimal revenue loss from the second private algorithm is upper bounded by $C_0\beta\epsilon_{\text{PTR}}$, where $\beta$ is the parameter used by the DP quantile for unbounded distribution.

The final ingredient is how the empirical reserve algorithm works for distribution $\mathcal{D}_p$; from Lemma I.6, we know that when $n = \theta(\epsilon^{-2}\log(\epsilon^{-1}))$, this revenue is at least $(1-\epsilon)\mathsf{OPT}(\mathcal{D}_p)$, i.e.,

$$|\mathsf{Rev}(M_{\text{ER}}, \mathcal{D}_p) - \mathsf{OPT}(\mathcal{D}_p)| \leq \epsilon \cdot \mathsf{OPT}(\mathcal{D}_p).$$

Hence, assuming $\delta_1 = \delta/4, \delta_2 = \delta/2$ and noticing that $\widehat{v}_{\max}$ still exceeds the optimal quantile for distribution $\mathcal{D}_p$, we can upper bound the expected revenue gap between empirical reserves on the private distribution. The optimal revenue of the original distribution becomes:

$$|\mathbb{E}[\mathsf{Rev}(M_{\text{ER}(\mathcal{D}_p)}, \mathcal{D}_p) - \mathsf{OPT}(\mathcal{D})]| \leq |\mathsf{Rev}(M_{\text{ER}(\mathcal{D}_p)}, \mathcal{D}_p) - \mathsf{OPT}(\mathcal{D}_p)| + |\mathsf{OPT}(\mathcal{D}_p) - \mathsf{OPT}(\mathcal{D})|$$

$$\leq \epsilon \cdot \mathsf{OPT}(\mathcal{D}_p) + \epsilon_a + \Theta(\beta\epsilon_{\text{PTR}}) \leq \Theta(\epsilon + \epsilon_a + \epsilon_{\text{PTR}})$$

$$\leq \widetilde{O}(\frac{\epsilon^2}{\epsilon_p} + \epsilon + \epsilon_q + \epsilon_a),$$

where the second to last inequality following from $C$ is a constant, and the last inequality follows from hide the log factors. Since each of the $k$ distribution would contribute this amount to the revenue loss, our statement as an revenue error bound as $\widetilde{O}(k(\frac{\epsilon^2}{\epsilon_p} + \epsilon + \epsilon_q + \epsilon_a))$. $\qquad\square$

Notice that this lemma only guarantees that the expectation of $\mathsf{SREV}_i$ is close to the expected optimal revenue of $\mathcal{D}_i$. We still needs to prove that the $\mathsf{SREV}_i$ converges to its expectation quickly, hence is clost to the underlying expected optimal revenue.

**Theorem I.11** (Accuracy Guarantee of DPKOPT). *Let all parameters be the same as stated in Lemma I.10, we have that: with probability $1 - \delta$,*

$$|\mathsf{KREV} - \sum_{i\in[k]} \mathbb{E}[\mathsf{OPT}(\mathcal{D}_i)]| \leq \widetilde{\Theta}(k(\epsilon + \epsilon^2/\epsilon_p + \epsilon_q + \epsilon_a))$$

*Proof.* From the proofs of previous lemma, we notice that the distribution $\widehat{S}$ is upper bounded by $\widehat{v}_{\max}$, hence upper bounded by a $\beta C_0$. We denote $C_1 := \beta C_0$ here, and applies the Chernoff to upper bound how $\mathsf{SREV}_i$ might deviates from its expectation. With probability $1 - \delta_3$,

$$|\mathsf{SREV}_i - \mathbb{E}[\mathsf{Rev}(M^i_{\text{ER}(\mathcal{D}_p)}, \mathcal{D}^i_p)]| \leq \Theta(\sqrt{1/n \cdot \log(1/\delta_3)})$$

Plugging in $n = \epsilon^{-2}\log(\epsilon^{-1})$ gives us that, with probability $1 - \delta_3$,

$$\mathsf{SREV}_i - E[\mathsf{Rev}(M^i_{\text{ER}(\mathcal{D}_p)}, \mathcal{D}^i_p)] \leq \widetilde{\Theta}(\epsilon)$$

Thus, we have that, with probabilty $1 - k\delta_3$, we have:

$$\mathsf{KREV} = \sum_{i\in[k]} \mathsf{SREV}_i$$

$$\leq \sum_{i\in[k]} \mathbb{E}[\mathsf{Rev}(M^i_{\text{ER}(\mathcal{D}_p)}, \mathcal{D}^i_p)] + \widetilde{\Theta}(k\epsilon)$$

$$\leq \sum_{i\in[k]} \mathbb{E}[\mathsf{OPT}(\mathcal{D}_i)] + \widetilde{\Theta}(k(\epsilon + \epsilon^2/\epsilon_p + \epsilon_q + \epsilon_a))$$

At the same time, $\mathsf{KREV} \geq \sum_{i\in[k]} \mathbb{E}[\mathsf{OPT}(\mathcal{D}_i)] - \widetilde{\Theta}(k(\epsilon + \epsilon^2/\epsilon_p + \epsilon_q + \epsilon_a))$. Now, let $\delta_3 = \delta$, thus we have that with probability $1 - 2k\delta$,

$$|\mathsf{KREV} - \sum_{i\in[k]} \mathbb{E}[\mathsf{OPT}(\mathcal{D}_i)]| \leq \widetilde{\Theta}(k(\epsilon + \epsilon^2/\epsilon_p + \epsilon_q + \epsilon_a))$$

Now let $\delta$ in the statement be $1/2k$ of the $\delta$ applied in the proof gives us the desired results. $\qquad\square$

## I.5 DP MYERSON FOR UNBOUNDED DISTRIBUTION

---

**Algorithm 9** DP Myerson, Unbounded Distribution DPMYERU$(V, \epsilon_q, \epsilon_a, h, \epsilon_p)$

---

**Input:** $n$ samples $V = \{\mathbf{v}_1, \ldots, \mathbf{v}_n\}$, parameter $n_1$, quantile discretization $\epsilon_q$, additive discretization $\epsilon_a$, regularity parameter $\eta$, privacy parameter $\epsilon_p$, truncation parameter $\epsilon_t$

1: KREV $\leftarrow$ DPKOPT$(\{\mathbf{v}_1, \ldots, \mathbf{v}_{n_1}\}, \epsilon_q, \epsilon_a, \epsilon_p, \eta)$.
  ▷ Use $n_1$ samples to get a $k$-approximation of the optimal revenue.
2: Truncation all remaining samples by $1/\epsilon_t \cdot$ KREV.
3: Discretize all the values into multiples of $\epsilon_a$, let the resulting samples be $\widehat{V}$.
4: Prepare the quantile to be estimated: $Q \leftarrow \{\epsilon_q, 2\epsilon_q, \ldots, \ldots, \lfloor (1/\epsilon_q) \rfloor \cdot \epsilon_q, 1\}$
5: For each dimension $d$, decide the prices on remaining samples:
  $\widehat{S}_{[d,:]} \leftarrow$ QESTIMATE$(Q, V_{[d,n_1:]}, \epsilon_p)$
  ▷ Apply DP quantile estimate on the discretized value space(Alg. 4).
6: Fit Myerson's mechanism as if the valuations is in $\widehat{S}$, each associated with probability $\epsilon_q$.

---

Intergrating the bound of the DPKOPT(Alg. 2) into Alg. 9 gives us the following accuracy bound:

**Theorem I.12** (Revenue Guarantee of Private Myerson, Unbounded (Alg. 9))**.** *Given $\epsilon \in [0, 1/4]$, $n$ samples $\widehat{V}$ of the joint distribution $\mathbf{D} \in [0, h]^k$, the output of Myerson fitted under* DPMYERU *(Alg. 9) is $(2k\epsilon_p, 0)$-DP, and the expected revenue of this mechanism is close to the optimal revenue of distribution $\mathbf{D}$, i.e., for $\epsilon_a = \epsilon_q = \epsilon, n = \Theta(\epsilon^2 \log(k/\delta))$ and $n_1 = \epsilon^{-2}\log(\epsilon^{-1})$, with probability $1 - \delta$,*

$$| \mathbb{E}[\mathsf{Rev}(M_{\text{DPMYERU}}, \mathbf{D}) - \mathsf{Rev}(M_{\mathbf{D}}, \mathbf{D})]| \leq \widetilde{O}(\epsilon_t + {k^2\epsilon}/{\epsilon_t} + {k^2\epsilon^2}/{\epsilon_p \epsilon_t})$$

*Furthermore, when $\epsilon_t = \sqrt{\epsilon}$, the bounds simplifies to:*

$$| \mathbb{E}[\mathsf{Rev}(M_{\text{DPMYERU}}, \mathbf{D}) - \mathsf{Rev}(M_{\mathbf{D}}, \mathbf{D})]| \leq \widetilde{O}(k^2\sqrt{\epsilon} + {k^2\epsilon^{1.5}}/{\epsilon_p})$$

*Proof.* From Thm H.4, we get that from $n = \widetilde{\Theta}(\epsilon^{-2})$ samples, with probability $1 - \delta$ and $\epsilon_a = \epsilon_q = \epsilon$ we get that:

$$| \mathbb{E}[\mathsf{Rev}(M_{\text{DPMYER}}, \mathbf{D}) - \mathsf{OPT}(\mathbf{D})]| \leq \widetilde{\Theta}((\epsilon + \epsilon^2/\epsilon_p)kh).$$

Next, we upper bound the value of the truncation point, from Thm I.11, with probability $1 - \delta$, we get:

$$|\text{KREV} - \sum_{i \in [k]} \mathbb{E}[\mathsf{OPT}(\mathcal{D}_i)]| \leq \widetilde{\Theta}(k(\epsilon + \epsilon^2/\epsilon_p))$$

Then, we have:

$$\text{KREV} \leq \sum_{i \in [k]} \mathbb{E}[\mathsf{OPT}(\mathcal{D}_i)] + \widetilde{\Theta}(k(\epsilon + \epsilon^2/\epsilon_p + \epsilon_q + \epsilon_a))$$
$$\leq k\mathsf{OPT}(\mathbf{D}) + \widetilde{\Theta}(k(\epsilon + \epsilon^2/\epsilon_p + \epsilon_q + \epsilon_a))$$
$$= \widetilde{\Theta}(k(1 + \epsilon + \epsilon^2/\epsilon_p))$$

Now we plug in $h = \frac{1}{\epsilon_t}$KREV, after simplification, this will gives us that, with probability $1 - 2\delta$:

$$| \mathbb{E}[\mathsf{Rev}(M_{\text{DPMYERU}}, \mathbf{D}) - \mathsf{Rev}(M_{\mathbf{D}}, \mathbf{D})]| \leq \widetilde{O}(\epsilon_t + {k^2\epsilon}/{\epsilon_t} + {k^2\epsilon^2}/{\epsilon_p \epsilon_t})$$

Now let the $\delta$ in the statement be the $1/2$ of the $\delta$ applied in this proof gives us the desired results.

$\square$

Since our DP Myerson for unbounded distribution integrates the DP Myerson for bounded distribution, and the estimation for optimal revenue at most take $O(n)$ pass of the whole distribution, we have the running time guarantee below. Notice that here the $\epsilon_t$ doesn't affects the running time

**Theorem I.13** (Running time, DP Myerson for Unbounded Distribution). *Given $n$ samples, and quantile discretization parameter $\epsilon_q$, the running time of* DPMYER *(Alg.9) is* $\Theta(k\log(1/\epsilon_q)n) = \widetilde{\Theta}(kn)$ *and requires* $\widetilde{O}(n)$ *pass of the distribution.*

*Proof.* From previous running time analysis of the DP Myerson for bounded distribution(Theorem H.5), we get that this part will take $\widetilde{\Theta}(1)$ pass of the distribution and has running time $\widetilde{\Theta}(kn)$.

The major component that contributes to the time complexity is the DP Quantile Estimation for Unbounded distribution, which, in the worst case, will take $O(n)$ passes over the dataset Durfee (2023), with each subsequent query takes $O(1)$ time for each distribution. Thus, the running time of DPQE for unbounded distribution is $O(kn)$.

Summing these up gives us a total running time of $\widetilde{\Theta}(kn)$.

$\square$

## J    MORE DETAILS FOR APPLICATION TO ONLINE MECHANISM DESIGN

### J.1    COMMITMENT MECHANISM

---
**Algorithm 10** Commitment Mechanism for bounded distribution

---
**Input:** Bids $\mathbf{b} \in \mathbb{R}_+^n$, distribution upper bound $h$.
  1: Sample a price $p \in [0, h]$ uniformly at random.
  2: Sample a bidder $i \in [n]$ uniformly at random.
  3: **if** $b_i > p$ **then** allocate the item to bidder $i$ with price $p$.
  4: **else** No bidder gets the item.
  5: **end if**

---

Our commitment algorithm (Alg. 10) selects each bidder with equal probability, with a price drawn uniformly from $[0, h]$.

### J.2    ONLINE MECHANISM DESIGN PRELIMINARIES

**Assumption J.1** (Bidders' Distributions). We assume there are $k$ publicly available bidder attributes, corresponding to $k$ different distributions, i.e., each bidder with the attribute $a \in [k]$ will sample their valuations from $\mathcal{D}_a$. These distributions are unknown to the learner (i.e., prior independent). At every iteration $t$, one bidder from each attribute participates in the auction, sees the item, and decides their valuations.[13] In addition, these valuations are independent across different bidders and rounds.

**Definition J.2** (Bidder's Utility). Each bidder $j$ has a quasi-linear utility function at time $t$: $u_j^t = x_j^t(v_j^t - p_j^t)$In our paper, we consider the following bidder models:

- **Discounted Utility**. For some discount factor $\gamma \in [0, 1]$, all bidders discount future utility by $\gamma$ and seek to maximize the sum of discounted utilities. At the $t$-th iteration, the discounted utility is $\widehat{u}_j^t = \sum_{r=t}^T u_j^r \gamma^{r-t}$.

- **Large Market** (Anari et al., 2014; Jalaly Khalilabadi and Tardos, 2018; Chen et al., 2016): $u_j^{1:T} = \sum_{t \in S_j} u_j^t$, with $|S_j| < l$.

where $S_j$ is the set of iterations that the bidder participates in for the auction and $x_j^t, v_j^t, p_j^t$ is the allocation, value, price for bidder $j$ at time $t$, respectively.

This assumption is essential to optimize a near-optimal revenue since it is impossible to obtain more than a constant fraction of the revenue in a single bidder setting if each bidder participates in every round of the mechanism (Lem D.1). Ideally, the learner's objective is to learn a revenue-maximizing auction with a small failure probability. This regret is comparable to the revenue of the best fixed mechanism against the (nonobservable) value history; hence, it is stronger than the traditional regret, which is comparable to the revenue of the best-fixed mechanism against the bid history.

**Definition J.3** (Learner's Objective). Given $\delta$, the goal of the learner is to decide an allocation $\mathbf{x}_{1:T}$ and $\mathbf{p}_{1:T}$ such that the cumulative revenue is near optimal and with sublinear regret, i.e., with probability $1 - \delta$:

$$\text{REGRET} := \frac{1}{T} \sum_{t \in [T]} \mathbb{E}[\text{Rev}(\mathbf{x}_t, \mathbf{p}_t, \mathbf{b}_t) - \mathbb{E}[\text{OPT}(\mathbf{v}_t)]] = o(1),$$

where the expectation is taken over the bidders' distribution.

### J.3    REVENUE GUARANTEE

**Theorem J.4** (Accuracy Guarantee of Two-stage Mechanism). *Given* $\epsilon \in [0, 1/4]$*,* $n$ *samples of the joint distribution* $\mathbf{D} \in [0, h]^k$*, let Alg. 3 run with parameter* $T_1 = \Theta(\epsilon^{-2}\log(k/\delta)), T = \Omega(T_1), \epsilon_a = \epsilon_q = \epsilon_p = \epsilon, \nu = 2\alpha$ *as calculated by Lemma J.7. Then, with probability* $1 - \delta$*, the regret is upper bounded, i.e.,*

---
[13]We assume the bidder cannot see her valuation of all items at the start of the process.

- **Large Market Bidder**: REGRET $= \widetilde{\Theta}[(\epsilon + \sqrt{l\epsilon})kh]$

- **Discounting Bidder**: REGRET $= \widetilde{\Theta}[(\epsilon + \sqrt{\frac{\gamma\epsilon}{1-\gamma}})kh]$.

*Proof.* First note that each mechanism in the rounds $T_1$ to $T$ is $(2k\epsilon_p, 0)$-DP. From Lemma J.7, we get that by our choice of $\nu$, during the first stage of our mechanism, each bid will at most deviate by $\nu$. Then, by Lemma J.9, the expected utility of the Myerson deployed in the second stage (without DP) satisfies that

$$|\mathbb{E}[\text{Rev}(M_{\widehat{\mathbf{D}}}, \mathbf{D}) - \text{Rev}(M_{\mathbf{D}}, \mathbf{D})]| \leq 2\nu,$$

where $\widehat{\mathbf{D}}$ is the distribution after subtracting in line 4 of Alg. 3. Then, from the proofs of the DP mechanism for bounded distribution (Thm. 3.2), we have that with probability $1 - \delta$, the expected utility of DPMYER on distribution $\widehat{\mathbf{D}}$ will have the following guarantee under $n = \Theta(\epsilon^{-2}\log(k/\delta))$:

$$|\mathbb{E}[\text{Rev}(M_{\widehat{P}_p}, \widehat{\mathbf{D}}) - \text{Rev}(M_{\widehat{\mathbf{D}}}, \widehat{\mathbf{D}})]| \leq \widetilde{\Theta}[(\epsilon + \epsilon^2/\epsilon_p)kh],$$

where $M_{\widehat{P}_p}$ denotes the algorithm applied in the second stage of our mechanism. Aggregating this guarantee gives us:

$$|\mathbb{E}[\text{Rev}(M_{\widehat{P}_p}, \mathbf{D}) - \text{Rev}(M_{\mathbf{D}}, \mathbf{D})]|$$
$$\leq |\mathbb{E}[\text{Rev}(M_{\widehat{P}_p}, \mathbf{D}) - \text{OPT}(\widehat{\mathbf{D}})]| + |\mathbb{E}[\text{OPT}(\widehat{\mathbf{D}}) - \text{Rev}(M_{\mathbf{D}}, \mathbf{D})]|$$
$$\leq \widetilde{\Theta}((\epsilon + \epsilon^2/\epsilon_p)kh + \nu).$$

We can now analyze the entire regret of our 2-stage algorithm. With probability $1 - \delta$,

$$\text{REGRET} = \mathbb{E}[\frac{T - T_1}{T}(\text{OPT}(\mathbf{D}) - \widetilde{\Theta}((\epsilon + \epsilon^2/\epsilon_p)kh + \nu)) - \text{OPT}(\mathbf{D})].$$

Plugging in the $\nu$ for the large market and the discounting bidder gives us the desired result. $\square$

### J.4 PROOF OF LEMMAS

**Lemma J.5** (Approximate Truthfulness via DP (Lemma 3 in McSherry and Talwar (2007))). *Given a mechanism $M$ with $(\epsilon_p, 0)$-DP and database $V, \widehat{V}$ that differs only in one entry, for any nonnegative function $g$, we have that*

$$\mathbb{E}[g(M(V))] \leq \exp(\epsilon_p) \cdot \mathbb{E}[g(M(\widehat{V}))].$$

*Moreover, this implies that $\mathbb{E}[g(M(V))] \in [(1 \pm 2\epsilon_p)\mathbb{E}[g(M(\widehat{V}))]]$.*

We now give a lemma controlling the distance between a bid and its value.

**Lemma J.6** (Bid Utility). *Given $\alpha \in [0, 1]$ and $t \in [0, T_1]$, the current utility at the $t$-th round of truthful bidding $v_t$ exceeds that of strategic bidding $b_t$ such that $|b_t - v_t| > 2\alpha$ by at least $\frac{2\alpha^2}{kh}$.*

*Proof.* WLOG, we assume $b_t < v_t - 2\alpha$. Then, the utility loss of strategic bidding is calculated as follows:

$$\int_{p=v_t-2\alpha}^{v_t} (v_t - p)\frac{1}{kh} = \frac{\alpha v_t}{kh} + \frac{2\alpha^2}{kh} \geq \frac{2\alpha^2}{kh}.$$

$\square$

**Lemma J.7** (Bid Deviation). *Given any $t \in [0, T_1]$, the bidder will bid only $b_t$ such that $|b_t - v_t| \leq 2\alpha$ for:*

- **Large Market**: $\sqrt{2(l-1)\epsilon_p}hk$.

- **Discounting Bidder**: $\sqrt{\frac{2\gamma\epsilon_p}{1-\gamma}}kh$.

*Proof.* The future utility of a bidder is upper bounded by: (1) $(l-1)h$ for the large market and (2) $h[1 + \frac{1}{\gamma} + \frac{1}{\gamma^2} + \ldots] = \frac{\gamma}{1-\gamma}h$ for the discounting bidder. Notice that for each round in $[T_1, T]$, the mechanism is $2k\epsilon_p$-DP; then, from Lemma J.5, we get that the future utility of strategic bidding is upper bounded by: (1) $4(l-1)hk\epsilon_p$ for the large market and (2) $\frac{\gamma}{1-\gamma}4kh\epsilon_p$ for the discounting bidder. Letting the current utility loss (i.e., $\frac{2\alpha^2}{kh}$) exceed the future rounds gives us a lower bound on $\alpha$: (1) $\sqrt{2(l-1)\epsilon_p}hk$ for the large market and (2) $\sqrt{\frac{2\gamma\epsilon_p}{1-\gamma}}kh$ for the discounting bidder. $\square$

We now present an auxiliary lemma that bounds the revenue gap between the optimal mechanism under $\mathcal{D}$ versus $\mathcal{D} - 2\nu$.

**Lemma J.8** (Loss under additive $\nu$). *Given a product distribution* $\mathbf{D} \in \mathbb{R}_+^k$*, let* $\widehat{\mathbf{D}} := \mathbb{P}_{\mathbb{R}+}[\mathbf{D} - \nu]$*, which results from subtracting $\nu$ for each value in $\mathbf{D}$ then projected onto positive value domain. Then, we have that*

$$\mathbb{E}[\mathsf{OPT}(\mathbf{D}) - \mathsf{OPT}(\widehat{\mathbf{D}})] \leq \nu.$$

*Proof.* Let $\widetilde{D} := \mathbf{D} - \nu$, then we couple distribution $\mathbf{D}$ and $\widetilde{D}$ such that each $v \in \mathbb{R}_+^k$ in $\mathbf{D}$ corresponds to $v - \gamma$ in $\widetilde{D}$, where $\mathbf{1}_k$ denotes the all one vector in $k$-dimensional space.

Then, we construct mechanism $M$ according to the optimal mechanism $M_\mathbf{D}$ as follows: For each value profile $v$, $x_M(v) = x_{M_\mathbf{D}}(v + \nu\mathbf{1}_k)$ and $p_M(v) = p_{M_\mathbf{D}}(v + \nu\mathbf{1}_k) - \mathbf{1}_k\nu$. Since $M_{\widetilde{D}}$ is monotonic (hence, truthful), $M$ is also truthful. Since all virtual value shifts equally, the allocation of $M$ corresponds to the allocation of the optimal mechanism for $\widetilde{D}$. Hence, we have

$$\mathbb{E}[\mathsf{OPT}(\mathbf{D}) - \mathsf{OPT}(\widehat{\mathbf{D}})] \geq \mathbb{E}[\mathsf{OPT}(\mathbf{D}) - \mathsf{OPT}(\widetilde{D})] + \mathbb{E}[\mathsf{OPT}(\widetilde{D}) - \mathsf{OPT}(\widehat{\mathbf{D}})]$$
$$\geq \mathbb{E}[\mathsf{Rev}(M_\mathbf{D}, \mathbf{D}) - \mathsf{Rev}(M_{\widehat{\mathbf{D}}}, \widehat{\mathbf{D}})]$$
$$\geq \mathbb{E}[\mathsf{Rev}(M_\mathbf{D}, \mathbf{D}) - \mathsf{Rev}(M, \widehat{\mathbf{D}})] = \nu,$$

where the first line follows from the definition of the optimal mechanism. $\square$

We can now give our bound on the revenue deviation due to untruthful bidding.

**Lemma J.9** (Revenue Deviation). *Given distribution $\mathbf{D}$, for any (bid) distribution $\widetilde{\mathbf{D}}$ resulting from perturbing at most $\pm\nu$ for each value of $\mathbf{D}$ and any distribution $\widehat{\mathbf{D}}$ resulting from subtracting $\gamma$ for each value of $\widetilde{\mathbf{D}}$, we have that*

$$0 \geq \mathbb{E}[\mathsf{Rev}(M_{\widehat{\mathbf{D}}}, \mathbf{D}) - \mathsf{Rev}(M_\mathbf{D}, \mathbf{D})] \geq -2\nu.$$

*Proof.* We clearly see that $\widehat{\mathbf{D}}$ satisfies $\mathbf{D} \succeq \widehat{\mathbf{D}}$. Since $M_\mathbf{D}$ is optimal for $\mathbf{D}$, we have $0 \geq \mathbb{E}[\mathsf{Rev}(M_{\widehat{\mathbf{D}}}, \mathbf{D}) - \mathsf{Rev}(M_\mathbf{D}, \mathbf{D})]$. It remains to give a lower bound of this revenue difference. By strong monotonicity (Lemma F.3), we have

$$\mathbb{E}[\mathsf{Rev}(M_{\widehat{\mathbf{D}}}, \mathbf{D}) - \mathsf{Rev}(M_\mathbf{D}, \mathbf{D})] \geq \mathsf{Rev}(M_{\widehat{\mathbf{D}}}, \widehat{\mathbf{D}}) - \mathsf{Rev}(M_\mathbf{D}, \mathbf{D}) \geq -2\nu,$$

where the last inequality follows from Lemma J.8.

$\square$