# OpenReview forum: "Private Mechanism Design via Quantile Estimation"
_ICLR.cc/2025/Conference — ICLR 2025 Poster_

### Official Review · Reviewer_377R · 2024-10-27

**Soundness:** 3
**Presentation:** 3
**Contribution:** 3
**Rating:** 6
**Confidence:** 3

**Summary:**

This paper combines differential privacy with auction theory, specifically focusing on revenue-maximizing auctions under pure DP.  The authors propose frameworks that handle independent, non-identical, and both bounded and unbounded valuation distributions.

**Strengths:**

The paper provides auctions for unbounded distributions and establishes theoretical guarantees.

**Weaknesses:**

1. There is no experiment.
2. I have some concerns about motivation and novelty, see Questions part for details.
3. The notation is confusing like different $\epsilon$ and in Theorem 5.4 , "the regret is upper bounded by $\tilde{\Theta}$" where $\tilde{\Theta}$ should be $\tilde{O}$.

**Questions:**

1. In the multiple-dimension case, it is well-known that DP community considers approximate DP since it can help improve $\sqrt{d}$ factor in utility where $d$ is the dimension like in [1]. In this paper, the authors consider $k$-dimension but achieve pure DP. The motivation is weird. What is the advantage of pure DP compared with approximate DP in your setting?
2. One should consider composition theorem for multiple rounds like in algorithm 3 to achieve central DP. Also, in lines 277-278, the final privacy guarantee should be $\epsilon$-DP and then divide it to each dimension.
3. I have some concerns about the novelty of the technical part since the paper just invokes and combines previous methods in other work.


[1].Daniel Kifer, Adam Smith, and Abhradeep Thakurta. Private convex empirical risk minimization and
high-dimensional regression. In Conference on Learning Theory, pages 25.1–25.40, 2012.

---

> ### Author Response · Authors · 2024-11-27
>
> We thank reviewer 377R for recognizing the soundness and presentation of this paper, and several comments/suggestions regarding DP technicals. We've **fixed all the notation issues** you've mentioned. We'd like to emphasize and clarify that in online auctions, **our use of differential privacy (DP) is guided by a different intuition** compared to its application in other DP-focused machine learning literature, and **adaptive composition over the total time horizon $T$ (hence approximate DP) is not necessary** in our online auction setting. Below, we address your comment/question in detail:
>
> **Q1**: There is no experiment.
>
> **A1**: Thank you for your suggestion! We have **added empirical experiments on DP Myerson across diverse distribution profiles**, demonstrating its significant advantage over the second-price auction. For details, please see the “Updates and Revisions in Response to Reviewer Feedback.”
>
> **Q2**:  In this paper, the authors consider **k-dimension but achieve pure DP**. The motivation is weird. What is the advantage of pure DP compared with approximate DP in your setting?
>
> **A2**: That's a good question. **Pure DP enables a direct upper bound on the future utility gain of any strategic bids**. It’s slightly unclear whether you’re referring to **the number of rounds $T$ or the number of distributions $k$** as the “dimension” in question. Thus, we will address each of them separately:
>
> >  **Advanced composition over $T$ rounds is unnecessary** in our setting as our analysis focuses on leveraging **per-auction privacy guarantees to address incentive issues**, rather than ensuring privacy for the entire online algorithm.
>
> - We kindly reiterate that **the privacy guarantee of any single-shot DP Myerson, rather than global privacy over $T$ rounds, is sufficient to ensure the approximate truthfulness of the bids**. Intuitively, in our online auction framework, bidders are incentivized to strategically bid only in the first stage to achieve a favorable outcome in the second stage. Importantly, at every iteration, a misreport by a bidder corresponds to a single entry change in the database. This setup allows us to **apply the guarantees of pure differential privacy (Lemma J.5) to upper bound the strategic utility of misreporting** in the current round. By telescoping across rounds, we can further upper bound the overall strategic utility over the entire auction process.
>
> - In contrast, for unbounded distributions, **approximate differential privacy fails to provide similarly reasonable upper bounds** to effectively regulate strategic behavior, in that their upper bounds on strategic utility scale with the distribution upper bound $h$.
>
> > **The privacy guarantee’s dependency on $k$ is negligible, so it's unnecessary to adaptive compose across $k$,** since $k$ is usually a small constant in our setting.
>
> - The privacy guarantee’s dependency on $k$ versus $\sqrt{k}$ is negligible, given that the number of types $k$ is typically small and constant (e.g., 6 for Meta ad auctions). While adaptive composition could improve privacy through an approximate DP guarantee, it would compromise incentive guarantees, as previously noted.
>
> - In addition, **applying a weaker privacy notion could eliminate the privacy dependency on $k$**. More specifically, applying joint differential privacy could kill the privacy dependency on $k$ under the same parameters as in our theorem statement, please see Appendix B.1 for more details.
>
> > **We’d also like to mention that pure DP is still desired and an important design target in online learning settings.**
>
> There is other literature ( i.e.,  [1], [2]) in online settings where pure DP is still desired and an important design target. In online learning, approximate privacy is often favored due to the difficulty of achieving pure privacy.
>
> **Q3**: One should consider the composition theorem for multiple rounds like in algorithm 3 to achieve central DP. Also, in lines 277-278, the final privacy guarantee should be $\epsilon$-DP and then divided it to each dimension.
>
> **A3**: Thank you for your thoughtful feedback. **We would like to kindly reiterate that we adopt a slightly non-standard approach to apply and treat the privacy guarantee.** In the context of the online auction application, the privacy guarantee of a single auction serves **as a tool to control bidders' incentives to misreport**, rather than to ensure that the entire pipeline satisfies a comprehensive privacy guarantee. Please see our response **A2** for more details.

---

> ### Author Response · Authors · 2024-11-27
>
> **Q4**: I have some concerns about the novelty of the technical part since the paper just invokes and combines previous methods in other work.
>
> **A4**: We respectfully have a different perspective on this point, as **our approach significantly extends prior methods in meaningful and non-trivial ways**. Specifically:
>
> - **Improvements over Previous QE**: **Previous work offers distribution-independent error bounds, but these bounds become less practical in our scenarios** where the number of samples from the underlying continuous distribution is large. In contrast, **our approach provides robust guarantees even in such cases**. Specifically, the accuracy bounds in previous work rely on sample features such as the minimum pairwise distance and the range of the dataset. As the minimum pairwise distance between samples increases, their QE error bounds degrade significantly. In contrast, our accuracy bound is derived under the assumption that the samples originate from an additively discretized distribution and are characterized by ks-distance. This assumption enables us to deliver a **tighter and more actionable accuracy bound**, specifically tailored to our requirements.
>
> - **(Approximate) Truthfulness through Pure Privacy**: **Existing approaches to controlling incentives for strategic bidding don’t apply to the unbounded distribution.**, as they rely on the approximate privacy guarantee $(\epsilon_p, \delta)$ of the auction to manage strategic utility. Specifically, in such works, the strategic utility is bounded above by a multiplicative factor of $\exp(\epsilon_p)$ and an additive term $\delta T h$, where $T$ is the total time horizon and $h$ is the upper bound of the distribution. On the contrary, **our approach is capable of handling this incentives for unbounded distributions, a limitation that previous works fail to address**.
>
>
>
> ### Minor Comment:
>
> **C1**:  In Theorem 5.4, "the regret is upper bounded by $\tilde{\Theta}$" where $\tilde{\Theta}$ should be $\tilde{O}$.
>
> **A1**: Nice catch, both $\Theta$ and $O$ actually work for our setting, **we unified the notations according to your request** in the revised version accordingly.
>
> **C2**: The notation is confusing like different $\epsilon$.
>
> **A2**:  We do indeed require three distinct $\epsilon$ values to correspond to the parameters used in different algorithms ($\epsilon_q$ for quantile estimation, $\epsilon_a$ for additive discretization and $\epsilon_p$ for privacy parameter). However, a notable property of our algorithm is that $ \epsilon_q$ and $\epsilon_a$ can be considered to be in the same asymptotic order, up to polylogarithmic factors.
>
> [1] Agarwal, N., & Singh, K. (2017, July). The price of differential privacy for online learning. In International Conference on Machine Learning (pp. 32-40). PMLR.
>
> [2] Jain, P., Raskhodnikova, S., Sivakumar, S., & Smith, A. (2023, July). The price of differential privacy under continual observation. In International Conference on Machine Learning (pp. 14654-14678). PMLR.

---

### Official Review · Reviewer_vD12 · 2024-11-04

**Soundness:** 4
**Presentation:** 3
**Contribution:** 3
**Rating:** 8
**Confidence:** 4

**Summary:**

This paper studies the problem of differentially private Myerson auction design under a Differential privacy constraint. The model is that there are k “types” of users and in each round, we get k buyers sampled from the distribution (assumed to be i.i.d.). Myerson’s foundational result defines the revenue maximizing truthful auction when the seller knows the distributions that the valuations are drawn from. In this work, the authors ask if we can privately learn enough about these distributions from samples to be able to run a near-optimal auction.

The main contribution is to show that indeed this is feasible. Using samples, one can learn the approximate quantiles of the distributions. Given the quantiles, the authors show that one can get a good approximation to the Myerson prices. In the case that the range of valuations is bounded, one can use any of the many known quantile estimation algorithms. In the case of unbounded range, one can use a recent result to estimate the range and then run quantile estimation.

The authors also show how for non-myopic bidders in a repeated game setting, one can use this private learning algorithm along with a commitment algorithm to get near optimal revenue.

**Strengths:**

- This paper expands the set of settings where DP can be used for mechanism design
- It combines DP tools with an understanding of auctions to allow learning from buyer’s bids while preserving incentive compatibility even with non-myopic bidders.

**Weaknesses:**

- No major weaknesses, perhaps with the exception of relavance to this conference. See questions below

**Questions:**

- It would be useful if the authors compared to the utility the exponential mechanism would offer in this case, ignoring computational constraints.
- It is also likely that the sampling from the continuous exp mech distribution can be done by being a little careful in understanding the distribution, especially since the structure of Myerson should allow you to independently sample each user’s price.
- The online mechanism designed in this work uses “greedy” by separating the explore and exploit phases. Can you do better by combining exploration and exploitation?

Nits:
- “Mechanism Design in the title seems too general for what the paper is doing. I would suggest putting Myerson Auction Design or at best Auction Design in the title.

---

> ### Author Response · Authors · 2024-11-27
>
> We sincerely thank Reviewer vD12 for recognizing the contribution, soundness, and novelty of our work. We greatly appreciate your effort in reviewing our paper. You have raised many **insightful questions and comments**, which we will address below.
>
> **Q1**: No major weaknesses, perhaps with the exception of relavance to this conference.
>
> **A1**: Thank you for acknowledging the strengths of our work and pointing out that it has no major weaknesses—**we truly appreciate your kind words**. However, we would like to respectfully emphasize that **our paper aligns with several subject areas of ICLR**, as we will elaborate below:
>
> - **Optimization**: Our work tackles the core challenge of maximizing auction revenue under privacy constraints, **a classic constrained optimization problem** that aligns directly with ICLR's focus on advanced optimization techniques in machine learning.
>
> - **Societal Considerations: Fairness, Safety, and Privacy**: Our work employs differential privacy to protect sensitive bidder information, such as valuations, aligning with **ICLR’s emphasis on privacy-preserving machine learning** and its societal impact. Its robust design guards against adversarial and strategic manipulation, ensuring **safer and more reliable** economic systems—particularly critical in applications like ad auctions and decentralized finance.
>
> - **General Machine Learning**: Our work applies machine learning concepts (i.e., differential privacy, quantile estimation) to auction theory, highlighting the **cross-disciplinary innovation valued by ICLR**. It shows how ML tools can tackle important real-world problems like ad auctions, a key application of online algorithms.
> In addition, we want to kindly argue that many works studying similar settings as ours are actually published in ML conferences similar to ICLR, for example, [1], [2], [3], etc.
>
> **Q2**: It would be useful if the authors compared the utility the exponential mechanism would offer in this case, ignoring computational constraints.
>
> **A2**: This is an excellent suggestion; however, implementing this version of the exponential mechanism is nearly **computationally intractable**. Furthermore, **deriving a closed-form expression for the resulting revenue guarantee is unclear** (even for the approximate version), which is another reason we **opted not to use the vanilla exponential mechanism**, as we will elaborate below.
>
> - (a) Without discretizing the value distribution or parameter space for the Myerson auction, **implementing the exact exponential mechanism is infeasible**, as there are infinitely many possible mechanisms (i.e., Myerson auctions) to choose from. The key challenge is that the (ironed) virtual value is a continuous function of the quantile of the underlying distribution, whose exact form is unknown—except that it is monotonically increasing with the value.
>
> - (b) **The approximate version of the exponential mechanism is computationally inefficient.** Let’s first describe the approximate mechanism, which we have also **revised and detailed in Appendix D.1**. We begin by additively or multiplicatively discretizing each distribution into a finite set of $ l $ valuations per dimension. Fitting a Myerson mechanism then becomes equivalent to determining an ordering of $kl$ possible valuations, where $k$ is the number of bidders or types. The number of possible orderings grows exponentially as $\Theta(k^{kl})$, and evaluating the utility of each ordering requires $O(kn)$ time. Consequently, the total computation time becomes exponential.
>
> - (c) **The revenue guarantee of the (approximate) exponential mechanism is distribution-dependent**, as the revenue from a given ordering varies with the distribution. Consequently, there is no reasonable closed-form lower bound for the normalization term of the sampling probabilities, which is essential for deriving the revenue guarantee of the exponential mechanism.
>
> **Q3**: It is also likely that the sampling from the continuous exp mech distribution can be done by being a little careful in understanding the distribution.
>
> **A3**: Thanks for the advice. if we only have sample access to the distribution, then **it is unclear how independently sampling each user’s price could help, since Myerson relies on virtual value curves that interact in complex ways across bidders**, making the sampling problem quite intricate, just as we stated in our response (c) in **A2**. **If you have any suggestions on how this could be approached, we would greatly appreciate hearing them!**

---

> > ### Author Response · Authors · 2024-11-27
> >
> > **Q4**: The online mechanism designed in this work uses “greedy” by separating the explore and exploit phases. Can you do better by combining exploration and exploitation?
> >
> > **A4**: Nice question. Alternatively, **it is possible to combine exploration and exploitation** by updating and deploying the DPMyerson mechanism at every iteration, accompanied by the same commitment mechanism with a small probability ( which depends on the total time horizon T). However, the guarantees provided by this approach would be **comparable to those of our proposed method**.
> >
> > In our paper, we adopt a two-stage approach because it requires **updating the distribution estimation (based on the underlying samples) only once**, making it more computationally efficient.
> >
> > [1]: Huh, J. S., & Kandasamy, K. (2024). Nash Incentive-compatible Online Mechanism Learning via Weakly Differentially Private Online Learning. ICML 2024
> >
> > [2]: Deng, Y., Mirrokni, V., & Zhang, H. (2022). Posted pricing and dynamic prior-independent mechanisms with value maximizers. Advances in Neural Information Processing Systems, 35, 24158-24169.
> >
> > [3]: Feng, Z., Liaw, C., & Zhou, Z. (2023, July). Improved online learning algorithms for ctr prediction in ad auctions. In International Conference on Machine Learning (pp. 9921-9937). PMLR.

---

### Official Review · Reviewer_QM6b · 2024-11-04

**Soundness:** 3
**Presentation:** 3
**Contribution:** 3
**Rating:** 6
**Confidence:** 4

**Summary:**

This paper studies how to learn the optimal single-item auction (Myerson auctions) from samples of non-iid bidders' values in a pure differentially private (pure DP) way. The authors reduce this problem to private quantile estimation of the bidders' value distributions. By doing that, they improve previous works in DP mechanism design in several aspects.  The authors also apply their results to online auction design with non-myopic bidders, obtaining a learning algorithm that is approximately truthful for the bidders and approximately revenue-optimal for the seller.

**Strengths:**

(S1) Previous attempts to integrate DP with prior-dependent auction design were all computationally inefficient or only guaranteed approximate DP.  This work is the first to achieve both computational efficiency and pure DP, which is a significant conceptual and technical contribution.

(S2) The authors show that their approach can be applied to online auction with non-myopic bidders, ensuring that bidders are approximately truthful and the seller can obtain approximately optimal revenue.  This generalizes the iid setting of Huang et al (2018a) to the non-iid setting, answering their open question. This is a novel contribution to the online auction design literature.

(S3) The private quantile estimation technique used in this work does not directly follow from previous work like Kaplan et al 2022.  The authors generalize previous techniques for distinct data points to the case with potentially identical data points.  This is an interesting contribution to the DP literature.

(S4) The writing is very good. For example, the results and contributions are clearly stated in the introduction. The proof sketches nicely summarize the main proof ideas.

**Weaknesses:**

(W1) The high level idea is not new. Huang et al (2018) and Abernethy (2019) have applied DP techniques to the problem of learning optimal auctions from non-myopic bidders. This paper is more of a refinement and generalization of previous works.

(W2) No experiments are given.  Some experimental results might strengthen the work a lot, given the practical motivation of using DP mechanism in real-world auctions.

Nevertheless, this work improves previous works in several aspects, like pure DP, non-identical bidders, unbounded distributions, and computational efficiency.  So I lean towards accepting this work.

**Questions:**

**Question:**

(Q1) Should the $V_{[i, :]}$ be $\hat V_{[i, :]}$ in Algorithm 1?  And what does $[i, :]$ mean?



**Suggestions:**

- Definition 2.1: The allocation rule and payment rule notations $\bf x$ and $\bf p$ should be vector-valued functions, not vectors.

- Typo in Definition 2.3: $exp$ $\epsilon$.

- Line 225: "increasing the sample size n from continuous value distributions naturally leads to points that are either very close or identical". Points sampled from a continuous distribution are identical with probability 0.

- The proof sketch of Theorem 3.2 and the formal proof (the proof of Theorem F.4 in the Appendix) didn't say why the QESTIMATE oracle is DP.  I guess it is given Lemma F.3.

---

> ### Author Response · Authors · 2024-11-27
>
> We sincerely thank Reviewer QM6b for your **detailed and thoughtful review**, and for recognizing the clarity, soundness, and novelty of our work. We have **carefully revised the manuscript in response to your valuable suggestions**. We’re happy to provide additional information or address any further questions upon request. Below, we address your comment/question in detail:
>
> ### Response to Questions:
>
> **Q1**: This paper is more of a refinement and generalization of previous works.
>
> **A1**: We would like to acknowledge that, while these related works share some similarities with ours—namely, their use of differential privacy to manage strategic behavior across rounds—there are **several key distinctions** that set our work apart:
>
> - The primary goal of our work is to develop **single-shot** approximate revenue-maximizing auctions with **pure** privacy guarantees, which are then applied to multi-shot online auctions. In contrast, **previous work has primarily focused on multi-round auctions** and relied on off-the-shelf solutions from online learning and differential privacy.
>
> - Our algorithm extends previous work to **unbounded distributions**, whereas earlier approaches handle only bounded ones. Huang et al. (2018) and Abernethy et al. (2019) use tree aggregation for DP internal states, but it’s unclear how to apply this to CDF proxies for unbounded distributions or maintain **pure** DP without significant revenue loss.
>
> **Q2**: No experiments are given.
>
> **A2**:  Thank you for your suggestion! We have **included empirical experiments on DP Myerson across various distribution profiles**, showing it significantly outperforms the second-price auction. For details, please refer to the “Updates and Revisions in Response to Reviewer Feedback.”
>
> **Q3**: Should the $V[i,:]$ be $V^[i,:]$ in Algorithm 1? And what does $[i,:]$ mean?
>
> **A3**: Nice catch. We construct $V$ as a dataset, where each row represents a bid type and each column corresponds to the index of a different sample. Specifically, $V[i,:]$ denotes all the samples for the $i$-th bid type.
>
> ### Response to Suggestions:
>
> **S1**:  The allocation rule and payment rule notations x and p should be vector-valued functions, not vectors.
>
> **A1**: Nice catch. We omitted them for simplicity but have now addressed this in the revised version.
>
> **S2**: Typo in Definition 2.3: $exp (\epsilon)$.
>
> **A2**: Nice catch! We have updated this expression in the revised version.
>
> **S3**: Line 225: "Increasing the sample size n from continuous value distributions naturally leads to points that are either very close or identical".
>
> **A3**: Thank you for pointing this out! We’ve addressed this in the updated version. To clarify, our intention was to convey that, **given a distance $\delta$, the probability of two points being closer than $\delta$ increases as the sample size grows**.
>
> **S4**: The proof sketch of Theorem 3.2 and the formal proof (Theorem F.4 in the Appendix) do not explain why the QESTIMATE oracle is DP; I assume this follows from Lemma F.3.
>
> **A4**: Nice catch! Yes, the privacy guarantee follows from Lemma F.3. **We’ve added more details on this in the revised version for clarity**.

---

> > ### Comment · Reviewer_QM6b · 2024-12-02
> >
> > Thank the authors for the great effort in the rebuttal!
> >
> > I do agree that refining previous work on DP mechanism design from approximate DP to pure DP and handling unbounded distributions are non-trivial theoretical contributions.
> >
> > Nevertheless, according to the added experimental results, the proposed mechanism does not seem to have very good empirical performance.  In Table 1 (Page 10) and other tables in Appendix A, DP Myerson's revenue is often closer to second price auction than to Myerson.  This is unsatisfactory because DP Myerson, as a DP version of Myerson, should ideally have a performance close to the non-DP Myerson auction.  This lets me doubt the practical performance of the pure DP mechanism proposed in this work.
> >
> > So I will maintain my relatively positive rating of 6, but not 8.

---

> ### Author Response · Authors · 2024-12-03
>
> Thank you for your timely response and insightful questions regarding the performance of DP Myerson. **We sincerely appreciate your thoughtful feedback!** We would like to further clarify your comments regarding the revenue gap. **The observed revenue gap stems from our choice of value profiles and hyperparameters rather than any suboptimality in DP Myerson**. Our current benchmark is assessed differently from a direct comparison:
>
> - **Objective in Comparing Myerson's Revenue** Our aim in comparing Myerson’s revenue is to assess whether **the revenue loss of DP Myerson aligns with theoretical expectations**. Unlike Myerson’s auction, DP Myerson incurs an additional revenue loss due to the quantile discretization step. The theoretical upper bound of this loss is 0.14, 0.45, and 0.2, which is higher than the actual revenue gaps observed between DP Myerson and Myerson—calculated as 0.07326, 0.12513, and 0.0738 for each respective distribution profile. These results experimentally demonstrate that the empirical revenue loss in DPQE can be attributed to the theoretically predicted impact of quantile discretization. For additional details, please refer to **Remark A.3** in our revised version.
>
> - **Comparison with Second-Price Auctions** Our goal in comparing DP Myerson with second-price auctions is to illustrate that DP Myerson performs effectively and significantly outperforms second-price auctions. Importantly, this **remains true even without fine-tuning hyperparameters**, further underscoring DP Myerson’s **robustness**.
>
> According to your insightful questions, we're currently working on **adding more comprehensive experiments**, including:
>
> - **Calculating the correct Myerson benchmark**—specifically, Myerson applied to both quantile-discretized and additively-discretized valuations.
>
> - Conducting similar experimental evaluations for **different value profiles**, which we will incorporate in future updates.
>
> **Thank you again for your valuable advice and feedback!**

---

### Official Review · Reviewer_Ncwf · 2024-11-04

**Soundness:** 3
**Presentation:** 3
**Contribution:** 3
**Rating:** 6
**Confidence:** 1

**Summary:**

The paper investigates the problem of designing differentially private (DP) mechanisms for single-item auctions that maximize revenue while ensuring ($\epsilon$, 0)-pure privacy. The authors propose two auction learning frameworks for both bounded and unbounded valuation distributions that privately estimate specific quantiles to achieve near-optimal revenue. Notably, the proposed frameworks maintain computational efficiency with only polylogarithmic overhead relative to non-private versions. Extending the approach to multi-round online auctions, the paper introduces mechanisms to limit strategic behavior from non-myopic bidders, demonstrating effectiveness in both bounded and unbounded distribution settings. The authors claim the method is the first efficient DP Myerson auction model under pure privacy and applicable to unbounded distributions.

**Strengths:**

The paper addresses a critical gap in differential privacy for revenue-maximizing auctions, especially in providing efficient algorithms that ensure pure DP, a stringent privacy model often overlooked in mechanism design. The focus on both bounded and unbounded distributions adds further novelty, and the integration with online auctions for non-myopic bidders is an innovative extension.

**Weaknesses:**

1. Experimental Validation: While the theoretical framework and proofs are robust, the paper lacks empirical validation. Demonstrating practical performance via simulation or real-world data could strengthen the results, showing real-world applicability of the bounded and unbounded auction mechanisms.

2. Complexity for Practitioners: Although theoretically sound, the use of pure DP and quantile estimation could be challenging for practitioners to implement in real auction settings. The paper might benefit from guidance on parameter selection or case studies showing how to configure the model in practical scenarios, particularly for choosing ε and handling non-i.i.d. bidder distributions.

3. Assumptions in Non-Myopic Bidder Models: The assumption of a commitment mechanism to prevent strategic bidding in online auctions may be limiting, as real-world non-myopic behavior could vary significantly. A sensitivity analysis on bidder behavior assumptions could improve robustness and applicability to broader settings.

**Questions:**

1. Experimental Validation: While the theoretical framework and proofs are robust, the paper lacks empirical validation. Demonstrating practical performance via simulation or real-world data could strengthen the results, showing real-world applicability of the bounded and unbounded auction mechanisms.

2. Complexity for Practitioners: Although theoretically sound, the use of pure DP and quantile estimation could be challenging for practitioners to implement in real auction settings. The paper might benefit from guidance on parameter selection or case studies showing how to configure the model in practical scenarios, particularly for choosing ε and handling non-i.i.d. bidder distributions.

3. Can the authors clarify how sensitive the revenue guarantee is to variations in the privacy parameter $\epsilon$? For instance, what level of revenue tradeoff might practitioners expect for specific values of $\epsilon$?

4. In the online auction application, how does the mechanism handle scenarios with highly irregular bidder participation, such as sporadic or bursty bidding behavior?

5. Could the approach be adapted or extended to multi-item auctions? If so, are there foreseeable challenges specific to quantile estimation or revenue guarantees?

6. What is the impact on revenue performance if the quantile estimation accuracy deviates from the bounds established in the paper? Would empirical validation potentially show scenarios where revenue significantly underperforms due to quantile estimation errors?

---

> ### Author Response · Authors · 2024-11-26
>
> We sincerely thank Reviewer Ncwf for recognizing the clarity, soundness, and novelty of our work. We greatly appreciate your effort in reviewing our paper. Below, we address your comment/question in detail:
>
> **Q1**: The paper lacks empirical validation.
>
> **A1**: Thank you for your suggestion. **We added experiments on different hyperparameters for DP Myerson** in the updated and revised version and also in the author rebuttal page, showing that it significantly **outperforms the second-price auction** across various distribution profiles. We are happy to include further experiments based on your recommendations.
>
> **Q2**: The paper could benefit from **guidance on parameter selection** or **case studies on practical model configuration**.
>
> **A2**: Thanks for the suggestion, we've add experiments for parameter selection in Appendix A.2
>
> - **Guidance on parameter**: In our main theorems regarding the revenue guarantee (i.e., Thm. 4.1 and 3.2), as long as $\epsilon_a$ and $\epsilon_q$ share the same asymptotic order (up to polynomial logarithmic factors), our theoretical near-optimal revenue guarantee remains valid.
> - **Case Studies on Practical Model Configuration**: In Appendix A.2, we present a case study investigating how different hyperparameters—additive discretization ($\epsilon_a$), quantile discretization ($\epsilon_q$), and the privacy parameter ($\epsilon_q$)—affect the revenue guarantee. Our findings reveal that **grid search is highly effective** for identifying optimal parameters that yield the best revenue guarantee across various distributions.
>
> **Q3**: Assuming a commitment mechanism to prevent strategic bidding in online auctions may be limiting, as real-world non-myopic behavior can vary widely.
>
> **A3**: This is an insightful comment! In fact, **with some milder and widely adopted alternative assumptions, our algorithm can still perform exceptionally well by replacing the commitment mechanism** with any prior-independent truthful auction (e.g., a second-price auction), as we elaborate below.
>
> - As stated in Appendix B.3, our algorithm guarantees sublinear regret under **Approximate Bayesian Incentive Compatible (BIC)** bidders [1], where bidders report truthfully as long as their future utility is upper-bounded by a small $\epsilon_B$. As demonstrated in the proof of Lemma J.7, BIC can be ensured by selecting an appropriate privacy parameter $\epsilon_p$, allowing the algorithm to maintain its sublinear regret guarantee.
>
> - Our algorithm guarantees sublinear regret under **bandit-based bidding strategies** (e.g., direct methods and inverse propensity score as in [2]), in that these strategies only depend on the history utility and don’t reason about future utilities, our algorithm still guarantees sublinear regret.
>
> **Q4**: A sensitivity analysis on bidder behavior assumptions could improve robustness and applicability to broader settings.
>
> **A4**: Thanks for the suggestion, as stated in our **A3**, our mechanism is **robust to other bidder behaviors**. We'd be glad to address any further related questions about specific bid behavior at the reviewer's request.
>
> **Q5**: Can the authors clarify how sensitive the revenue guarantee is to variations in the privacy parameter $\epsilon_p$?
>
> **A5**:
> - **Empirical Indications**. Our experiment results ( **Table 1,5,6** ) show that the revenue is **not sensitive** to variations in the privacy parameter $\epsilon_p$. **Privacy mechanisms appear to act as regularizers in this context**, effectively minimizing overfitting and enhancing generalization.
>
> - **Theoretical Analysis**. **$\epsilon_p$ significantly impacts the achievable revenue only when it is asymptotically smaller than either $\epsilon_a$ or $\epsilon_q$.** Take the private Myerson mechanism for bounded distributions (Thm. 3.2) for an example , for a given $(\epsilon_p, 0)$-DP output, the expected revenue is $\tilde{\Theta}(\epsilon + \epsilon^2 / \epsilon_p) k h$.
>
> - - When $\epsilon_p = \tilde{\Omega}(\epsilon)$, the revenue guarantee simplifies to $\tilde{\Theta}(\epsilon k h)$, where the dependence on $\epsilon_p$ is captured only through polylogarithmic factors hidden in $\tilde{\Theta}$.
>
> - - When $\epsilon_p = o(\epsilon)$, the revenue guarantee is dominated by the term $\epsilon^2 / \epsilon_p$.
>
> A minor note: **The commonly used privacy parameter in practical scenarios has little influence on the revenue guarantee.** More specifically, as indicated by Google's guidelines (Section 5.2 of [3]), empirical studies [4], and the U.S. Census Bureau [5], the privacy parameters are typically small constants (e.g., ~0.1). Consequently, $\epsilon_p$ is generally not as small as the accuracy parameter $\epsilon$, and the impact of $\epsilon_p$ on revenue, when expressed using big-$O$ notation, is insignificant.

---

> ### Author Response · Authors · 2024-11-26
>
> **Q6**: In the online auction application, how does the mechanism handle scenarios such as sporadic or bursty bidding behavior?
>
> **A6**: As stated in Appendix B.3, **our mechanism is robust to changes in the number of bidders, provided that we have a sufficient number of samples from the first stage.** The key lies in the accuracy of the virtual value learned from the first-stage output. As long as this virtual value estimate is reasonably accurate, it can still be used to construct a Myerson mechanism tailored to the current market conditions, given that the types of bidders are known.
>
> - **Our mechanism easily generalizes to handle sporadic behavior** because such behavior—where each bidder participates in only a small number of auction rounds—aligns naturally with the assumptions of our large market model.
> - **Fixes to bursty behavior**.  A possible modification is to allocate the item only when the number of bidders reaches a certain threshold. If the value distributions are (time)-independent, we can still apply a Myerson mechanism using the same fitted virtual value. This is an interesting future direction.
>
> **Q7**: Could the approach be adapted or extended to multi-item auctions? If so, are there foreseeable challenges specific to quantile estimation or revenue guarantees?
>
> **A7**: Nice question. Here are a few challenges for the generalization of our current framework to multi-item auctions:
> - **Degree of Freedom on multi-item auction is high**. Multi-item auctions offer significantly greater flexibility than single-item auctions, with more design choices and increased complexity. Revenue-maximizing auctions in this setting are particularly intricate, as bidders' utility functions for item sets can vary widely—unit-demand, submodular, additive, or even supermodular—each requiring distinct approaches to auction design.
> - **Implicit Revenue-Optimal Solution:** Furthermore, it remains unclear whether a clear, explicit form of revenue-maximizing auctions exists for multi-item settings. Without an explicit characterization of such solutions, it becomes challenging to theoretically analyze how our QE procedure impacts their accuracy guarantees.
>
> Nevertheless, **our QE framework can be adapted to the differential economic framework for multi-item auctions**. Specifically, this involves leveraging artificial neural network architectures designed to learn optimal solutions from sample data [6]. In this context, performance is evaluated empirically rather than theoretically.
>
> **Q8**: What is the impact on revenue performance if the quantile estimation accuracy deviates from the bounds established in the paper?
>
> **A8**: **This low-probability failure is an inherent and well-understood limitation in statistical machine learning.** With a small probability, our algorithm may fail to provide a reasonable revenue guarantee. However, this failure probability is both well-recognized and unavoidable in statistical machine learning, as the empirical distribution occasionally deviates significantly from the ground truth with low likelihood.
>
> **Q9**: Would empirical validation potentially show scenarios where revenue significantly underperforms due to quantile estimation errors?
>
> **A9**: Good question. **This phenomenon can occur with large and inappropraite $\epsilon_q$**, as noted in our initial empirical validation for certain distribution profiles, where the theoretical upper bound on revenue loss from quantile estimation $\epsilon_q h$ may be significant relative to the optimal revenue from the value distribution. **That said, this loss diminishes as $\epsilon_q$ decreases.** With hyperparameter choices consistent with our theorems, our algorithm still maintains **asymptotic optimality in revenue guarantees.**
>
>
> [1]: Balseiro, S. R., Besbes, O., & Castro, F. (2024). Mechanism design under approximate incentive compatibility. Operations Research, 72(1), 355-372.
>
> [2]: Jeunen, O., Murphy, S., & Allison, B. Learning to Bid with AuctionGym. AdKdd 2022.
>
> [3]: Ponomareva, N., Hazimeh, H., Kurakin, A., Xu, Z., Denison, C., McMahan, H. B., ... & Thakurta, A. G. (2023). How to dp-fy ml: A practical guide to machine learning with differential privacy. Journal of Artificial Intelligence Research, 77, 1113-1201.
>
> [4]: Nanayakkara, P., Smart, M. A., Cummings, R., Kaptchuk, G., & Redmiles, E. M. (2023). What are the chances? explaining the epsilon parameter in differential privacy. In 32nd USENIX Security Symposium (USENIX Security 23) (pp. 1613-1630).
>
> [5]: Abowd, J. M. (2018, July). The US Census Bureau adopts differential privacy. In Proceedings of the 24th ACM SIGKDD international conference on knowledge discovery & data mining (pp. 2867-2867).
>
> [6]: Dütting, P., Feng, Z., Narasimhan, H., Parkes, D. C., & Ravindranath, S. S. (2024). Optimal auctions through deep learning: Advances in differentiable economics. Journal of the ACM, 71(1), 1-53.

---

### Meta-Review · Area_Chair_zfnu · 2024-12-16

**Metareview:**

Reviewers liked the topic of private auction mechanism design, the simple mechanism, and the nice theoretical results. Most concerns were addressed in the rebuttal. One major concern was the lack of experiments. The authors added experiments, though some reviewer found them not as convincing as expected. Nevertheless, the consensus is that this (the lack of experiments) should not prevent the paper from being accepted.

**Additional Comments On Reviewer Discussion:**

See above

---

### Decision · Program_Chairs · 2025-01-22

Accept (Poster)